# Generalizing Denoising to Non-Equilibrium Structures Improves Equivariant Force Fields

**Yi-Lun Liao**                                                                        *ylliao@mit.edu*
*Massachusetts Institute of Technology*
*Work partially done during an internship at FAIR, Meta*

**Tess Smidt**                                                                         *tsmidt@mit.edu*
*Massachusetts Institute of Technology*

**Muhammed Shuaibi⋆**                                                                  *mshuaibi@meta.com*
*FAIR, Meta*

**Abhishek Das⋆**                                                                      *abhshkdz@gmail.com*
*Work done at FAIR, Meta*

⋆*Equal contribution*

**Reviewed on OpenReview:** https://openreview.net/forum?id=whGzYUbIWA

**Code:** https://github.com/atomicarchitects/DeNS

## Abstract

Understanding the interactions of atoms such as forces in 3D atomistic systems is fundamental to many applications like molecular dynamics and catalyst design. However, simulating these interactions requires compute-intensive *ab initio* calculations and thus results in limited data for training neural networks. In this paper, we propose to use **de**noising **n**on-equilibrium **s**tructures (**DeNS**) as an auxiliary task to better leverage training data and improve performance. For training with DeNS, we first corrupt a 3D structure by adding noise to its 3D coordinates and then predict the noise. Different from previous works on denoising, which are limited to equilibrium structures, the proposed method generalizes denoising to a much larger set of non-equilibrium structures. The main difference is that a non-equilibrium structure does not correspond to local energy minima and has non-zero forces, and therefore it can have many possible atomic positions compared to an equilibrium structure. This makes denoising non-equilibrium structures an ill-posed problem since the target of denoising is not uniquely defined. Our key insight is to additionally encode the forces of the original non-equilibrium structure to specify which non-equilibrium structure we are denoising. Concretely, given a corrupted non-equilibrium structure and the forces of the original one, we predict the non-equilibrium structure satisfying the input forces instead of any arbitrary structures. Since DeNS requires encoding forces, DeNS favors equivariant networks, which can easily incorporate forces and other higher-order tensors in node embeddings. We study the effectiveness of training equivariant networks with DeNS on OC20, OC22 and MD17 datasets and demonstrate that DeNS can achieve new state-of-the-art results on OC20 and OC22 and significantly improve training efficiency on MD17.

## 1 Introduction

Graph neural networks (GNNs) have made remarkable progress in approximating high-fidelity, compute-intensive quantum mechanical calculations like density functional theory (DFT) for atomistic systems (Gilmer

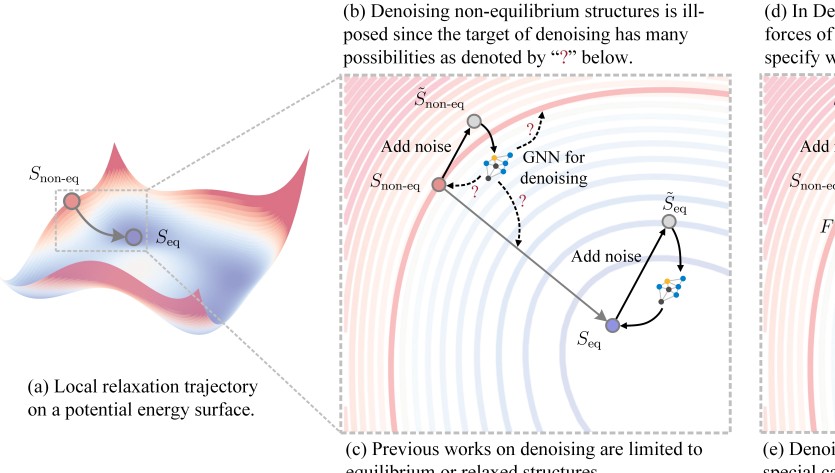

(b) Denoising non-equilibrium structures is ill-posed since the target of denoising has many possibilities as denoted by "?" below.

(d) In DeNS, we propose to additionally use the forces of the original non-equilibrium structure to specify which target structure we are denoising.

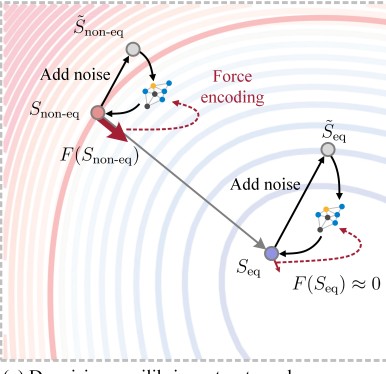

(a) Local relaxation trajectory on a potential energy surface.

(c) Previous works on denoising are limited to equilibrium or relaxed structures.

(e) Denoising equilibrium structures becomes a special case where forces are close to zero.

Figure 1: Illustration of denoising equilibrium and non-equilibrium structures. As shown in (a), we relax a non-equilibrium structure $S_{\text{non-eq}}$ and obtain the final relaxed, equilibrium structure $S_{\text{eq}}$. The path between $S_{\text{non-eq}}$ and $S_{\text{eq}}$ forms a relaxation trajectory. All structures $S$ along the trajectory except $S_{\text{eq}}$ are non-equilibrium and have non-zero forces $F(S)$. For denoising structures $S$, we add noise to their 3D atomic coordinates to obtain corrupted structures $\tilde{S}$ and predict the corresponding noise given $\tilde{S}$. We compare denoising non-equilibrium and equilibrium structures in (b) and (c), respectively, and show that denoising non-equilibrium structures can be ill-posed in (b). The issue in (b) can be addressed with force encoding, where we take forces as additional inputs, as in (d) and (e).

et al., 2017; Zhang et al., 2018; Unke et al., 2021; Batzner et al., 2022; Rackers et al., 2023; Lan et al., 2022), enabling new insights in applications such as molecular dynamics simulations (Musaelian et al., 2023) and catalyst design (Chanussot* et al., 2021; Lan et al., 2022). However, unlike other domains such as natural language processing (NLP) and computer vision (CV), the scale of atomistic data is quite limited since generating data requires compute-intensive *ab initio* calculations. For example, the largest atomistic dataset, OC20 (Chanussot* et al., 2021), contains about 138M examples while GPT-3 (Brown et al., 2020) is trained on hundreds of billions of words and ViT-22B (Dehghani et al., 2023) is trained on around 4B images.

To start addressing this gap, we take inspiration from self-supervised learning methods in NLP and CV and explore how we can adapt them to learn better atomistic representations from existing labeled data. Specifically, one of the most popular self-supervised learning methods in NLP (Devlin et al., 2019) and CV (He et al., 2022) is training a denoising autoencoder (Vincent et al., 2008), where the idea is to mask or corrupt a part of the input data and learn to reconstruct or denoise the original, uncorrupted data. Denoising assumes we know a unique target structure such as a sentence and an image in the case of NLP and CV. Indeed, this is the case for equilibrium structures (e.g., $S_{\text{eq}}$ at a local energy minimum in Figure 1(c)) as has been demonstrated by previous works leveraging denoising for pretraining on atomistic data (Jiao et al., 2022; Zaidi et al., 2023; Liu et al., 2023; Wang et al., 2023; Feng et al., 2023a). However, most previous works are limited to equilibrium structures, and equilibrium structures constitute only a small portion of available data since structures along a relaxation trajectory to get to a local minimum are all non-equilibrium as shown in Figure 1. Hence, it is important to generalize denoising to leverage the much larger set of non-equilibrium structures.

Since a non-equilibrium structure has non-zero atomic forces and atoms are not confined to local energy minima, it can have more possible atomic positions than a structure at equilibrium. As shown in Figure 1(b), this can make denoising a non-equilibrium structure an ill-posed problem since there are many possible target structures. To address the issue, we propose force encoding and take the forces of the original non-equilibrium structures as inputs when denoising non-equilibrium structures. Intuitively, the forces constrain the atomic positions of a non-equilibrium structure. With the additional information, we are able to predict the original non-equilibrium structure satisfying the input forces instead of predicting any arbitrary structures as shown in Figure 1(d). Previous works on denoising equilibrium structures (Godwin et al., 2022; Jiao et al., 2022;

Zaidi et al., 2023; Liu et al., 2023; Feng et al., 2023b;a) end up being a special case where the forces of original structures are close to zero as in Figure 1(e).

Based on the insight, in this paper, we propose to use **de**noising **n**on-equilibrium **s**tructures (**DeNS**) as an auxiliary task to better leverage atomistic data. For training DeNS, we first corrupt a structure by adding noise to its 3D atomic coordinates and then reconstruct the original uncorrupted structure by predicting the noise. For noise prediction, a model is given the forces of the original uncorrupted structure as inputs to make the transformation from a corrupted non-equilibrium structure to an uncorrupted non-equilibrium structure tractable. When used along with the original tasks like predicting the energy and forces of non-equilibrium structures, DeNS improves the performance on the original tasks with a marginal increase in training cost. We further discuss how DeNS can better leverage training data to improve performance and show the connection to self-supervised learning methods in other domains.

Because DeNS requires encoding forces, it favors equivariant networks, which build up equivariant features at each node with vector spaces of irreducible representations (irreps) and have interactions or message passing between nodes with equivariant operations like tensor products. Since forces can be projected to vector spaces of irreps with spherical harmonics, equivariant networks can easily incorporate forces in node embeddings. Moreover, with the reduced complexity of equivariant operations (Passaro & Zitnick, 2023) and incorporation of the Transformer network design (Liao & Smidt, 2023; Liao et al., 2023) from NLP (Vaswani et al., 2017) and CV (Dosovitskiy et al., 2021), equivariant networks have become the state-of-the-art methods on large-scale atomistic datasets.

We mainly focus on how DeNS can improve equivariant networks and conduct extensive experiments on OC20 (Chanussot* et al., 2021), OC22 (Tran* et al., 2022) and MD17 (Chmiela et al., 2017; Schütt et al., 2017; Chmiela et al., 2018) datasets. On OC20 S2EF-2M dataset, EquiformerV2 (Liao et al., 2023) trained with DeNS can achieve better energy and force results and save 2.3× training time compared to training without DeNS (Section 4.1.1). On OC20 S2EF-All+MD dataset, EquiformerV2 trained with DeNS achieves new state-of-the-art results on Structure to Energy and Forces (S2EF) and Initial Structure to Relaxed Energy (IS2RE) tasks (Section 4.1.2). Similarly, EquiformerV2 trained with DeNS sets new state-of-the-art results on OC22 dataset, improving energy by up to 15%, forces by up to 12%, and IS2RE by up to 15% (Section 4.2). On MD17 dataset, Equiformer ($L_{max} = 2$) (Liao & Smidt, 2023) trained with DeNS achieves better results and saves 3.1× training time compared to Equiformer ($L_{max} = 3$) without DeNS (Section 4.3), where $L_{max}$ denotes the maximum degree. Additionally, DeNS can improve other equivariant networks like eSCN (Passaro & Zitnick, 2023) on OC20 and SEGNN-like networks (Brandstetter et al., 2022) on MD17.

## 2 Related Works

**Denoising 3D Atomistic Structures.** Denoising structures have been used to boost the performance of GNNs on 3D atomistic datasets (Godwin et al., 2022; Jiao et al., 2022; Zaidi et al., 2023; Liu et al., 2023; Feng et al., 2023b; Wang et al., 2023; Feng et al., 2023a). The approach is to first corrupt data by adding noise and then train a denoising autoencoder to reconstruct the original data by predicting the noise, and the motivation is that learning to reconstruct data enables learning generalizable representations (Devlin et al., 2019; He et al., 2022; Godwin et al., 2022; Zaidi et al., 2023). Since denoising equilibrium structures do not require labels and is self-supervised, similar to BERT (Devlin et al., 2019) and MAE (He et al., 2022), it is common to pre-train via denoising on a large dataset of equilibrium structures like PCQM4Mv2 (Nakata & Shimazaki, 2017) and then fine-tune with supervised learning on smaller downstream datasets. Besides, the work of Noisy Nodes (Godwin et al., 2022) uses denoising equilibrium structures as an auxiliary task along with original tasks without pre-training on another larger dataset. However, most of the previous works are limited to equilibrium structures, which occupy a much smaller amount of data than non-equilibrium ones. In contrast, the proposed DeNS generalizes denoising to non-equilibrium structures with force encoding so that we can improve the performance on the larger set of non-equilibrium structures. We provide a detailed comparison to previous works on denoising in Section A.1.

**SE(3)/E(3)-Equivariant Networks.** Please refer to Section A.2 for discussion on equivariant networks. In addition, since we mainly focus on applying the proposed DeNS to Equiformer series (Liao & Smidt, 2023; Liao et al., 2023), we provide a brief introduction to them in Section A.3.

## 3 Method

### 3.1 Problem Setup

Calculating quantum mechanical properties like energy and forces of 3D atomistic systems is fundamental to many applications. An atomistic system can be one or more molecules, a crystalline material and so on. Specifically, each system $S$ is an example in a dataset and can be described as $S = \{(z_i, \mathbf{p}_i) \mid i \in \{1, ..., |S|\}\}$, where $z_i \in \mathbb{N}$ denotes the atomic number of the $i$-th atom and $\mathbf{p}_i \in \mathbb{R}^3$ denotes the 3D atomic position. The energy of $S$ is denoted as $E(S) \in \mathbb{R}$, and the atom-wise forces are denoted as $F(S) = \{\mathbf{f}_i \in \mathbb{R}^3 \mid i \in \{1, ..., |S|\}\}$, where $\mathbf{f}_i$ is the force acting on the $i$-th atom. In this paper, we define a system to be an equilibrium structure if all of its atom-wise forces are close to zero. Otherwise, we refer to it as a non-equilibrium structure. Since non-equilibrium structures have non-zero atomic forces and thus are not at an energy minimum, they have more degrees of freedom and constitute a much larger set of possible structures than those at equilibrium.

In this work, we focus on the task of predicting energy and forces given non-equilibrium structures. Specifically, given a non-equilibrium structure $S_{\text{non-eq}}$, GNNs predict energy $\hat{E}(S_{\text{non-eq}})$ and atom-wise forces $\hat{F}(S_{\text{non-eq}}) = \left\{\hat{\mathbf{f}}_i(S_{\text{non-eq}}) \in \mathbb{R}^3 \mid i \in \{1, ..., |S_{\text{non-eq}}|\}\right\}$ and minimize the following loss function:

$$\lambda_E \cdot \mathcal{L}_E + \lambda_F \cdot \mathcal{L}_F = \lambda_E \cdot |E'(S_{\text{non-eq}}) - \hat{E}(S_{\text{non-eq}})| + \lambda_F \cdot \frac{1}{|S_{\text{non-eq}}|} \sum_{i=1}^{|S_{\text{non-eq}}|} |\mathbf{f}'_i(S_{\text{non-eq}}) - \hat{\mathbf{f}}_i(S_{\text{non-eq}})|^2 \quad (1)$$

where $\lambda_E$ and $\lambda_F$ are energy and force coefficients controlling the relative importance between energy and force predictions. $E'(S_{\text{non-eq}}) = \frac{E(S_{\text{non-eq}}) - \mu_E}{\sigma_E}$ is the normalized ground truth energy obtained by first subtracting the original energy $E(S_{\text{non-eq}})$ by the mean of energy labels in the training set $\mu_E$ and then dividing by the standard deviation of energy labels $\sigma_E$. Similarly, $\mathbf{f}'_i = \frac{\mathbf{f}_i}{\sigma_F}$ is the normalized atom-wise force. For force prediction, we can either use direct methods, which is to directly predict forces from latent representations like node embeddings, as commonly used for OC20 and OC22 datasets or gradient methods, which is to take the negative gradients of predicted energy with respect to atomic positions, for datasets like MD17.

### 3.2 Denoising Non-Equilibrium Structures (DeNS)

#### 3.2.1 Formulation of Denoising

Denoising structures has been used to improve the performance of GNNs on 3D atomistic datasets (Godwin et al., 2022; Zaidi et al., 2023; Feng et al., 2023b; Wang et al., 2023). We first corrupt data by adding noise and then train a denoising autoencoder to reconstruct the original data by predicting the noise. Specifically, given a 3D atomistic system $S = \{(z_i, \mathbf{p}_i) \mid i \in \{1, ..., |S|\}\}$, we create a corrupted structure $\tilde{S}$ by adding Gaussian noise with a tuneable standard deviation $\sigma$ to the atomic positions $\mathbf{p}_i$ of the original structure $S$:

$$\tilde{S} = \{(z_i, \tilde{\mathbf{p}}_i) \mid i \in \{1, ..., |S|\}\}, \quad \text{where} \quad \tilde{\mathbf{p}}_i = \mathbf{p}_i + \boldsymbol{\epsilon}_i \quad \text{and} \quad \boldsymbol{\epsilon}_i \sim \mathcal{N}(0, \sigma I_3) \quad (2)$$

We denote the set of noise added to $S$ as $\text{Noise}(S, \tilde{S}) = \left\{\boldsymbol{\epsilon_i} \in \mathbb{R}^3 \mid i \in \{1, ..., |S|\}\right\}$. When training a denoising autoencoder, GNNs take $\tilde{S}$ as inputs, output atom-wise noise prediction $\hat{\boldsymbol{\epsilon}}(\tilde{S})_i$ and minimize the L2 difference between normalized noise $\frac{\boldsymbol{\epsilon}_i}{\sigma}$ and noise prediction $\hat{\boldsymbol{\epsilon}}(\tilde{S})_i$:

$$\mathbb{E}_{p(S, \tilde{S})} \left[ \frac{1}{|S|} \sum_{i=1}^{|S|} \left| \frac{\boldsymbol{\epsilon}_i}{\sigma} - \hat{\boldsymbol{\epsilon}}(\tilde{S})_i \right|^2 \right] \quad (3)$$

where $p(S, \tilde{S})$ denotes the probability of obtaining the corrupted structure $\tilde{S}$ from the original structure $S$. We divide the noise $\boldsymbol{\epsilon}_i$ by the standard deviation $\sigma$ to normalize the outputs of noise prediction.

When the original structure $S$ is an equilibrium structure, denoising is to find the structure corresponding to the nearest energy local minima given a high-energy corrupted structure. This makes denoising equilibrium structures a many-to-one mapping and therefore a well-defined problem. However, when the original structure $S$ is a non-equilibrium structure, denoising, the transformation from a corrupted non-equilibrium structure to the original non-equilibrium one, can be an ill-posed problem since there are many possible target structures as shown in Figure 1(b).

### 3.2.2 Force Encoding

The reason that denoising non-equilibrium structures can be ill-posed is because we do not provide sufficient information to specify the properties of the target structures. Concretely, given an original non-equilibrium structure $S_{\text{non-eq}}$ and its corrupted counterpart $\tilde{S}_{\text{non-eq}}$, some structures interpolated between $S_{\text{non-eq}}$ and $\tilde{S}_{\text{non-eq}}$ could be in the same data distribution and therefore be the potential target structures of denoising. In contrast, when denoising equilibrium structures as shown in Figure 1(c), we implicitly provide the extra information that the target structure should be at equilibrium with near-zero forces, and this therefore limits the possibility of the target of denoising. Motivated by the assumption that the forces of the original structures are close to zeros when denoising equilibrium ones, we propose to encode the forces of original non-equilibrium structures when denoising non-equilibrium ones as illustrated in Figure 1(d). Specifically, when training **de**noising **n**on-equilibrium **s**tructures (DeNS), GNNs take both a corrupted non-equilibrium structure $\tilde{S}_{\text{non-eq}}$ and the forces $F(S_{\text{non-eq}})$ of the original non-equilibrium structure $S_{\text{non-eq}}$ as inputs, output atom-wise noise prediction $\hat{\boldsymbol{\epsilon}}\left(\tilde{S}_{\text{non-eq}}, F(S_{\text{non-eq}})\right)_i$ and minimize the L2 difference between normalized noise $\frac{\boldsymbol{\epsilon}_i}{\sigma}$ and noise prediction $\hat{\boldsymbol{\epsilon}}\left(\tilde{S}_{\text{non-eq}}, F(S_{\text{non-eq}})\right)_i$:

$$\mathcal{L}_{\text{DeNS}} = \mathbb{E}_{p(S_{\text{non-eq}}, \tilde{S}_{\text{non-eq}})} \left[ \frac{1}{|S_{\text{non-eq}}|} \sum_{i=1}^{|S_{\text{non-eq}}|} \left| \frac{\boldsymbol{\epsilon}_i}{\sigma} - \hat{\boldsymbol{\epsilon}}\left(\tilde{S}_{\text{non-eq}}, F(S_{\text{non-eq}})\right)_i \right|^2 \right] \tag{4}$$

Equation 4 is more general and reduces to Equation 3 when the target of denoising becomes equilibrium structures with near-zero forces. Since we train GNNs with $\tilde{S}_{\text{non-eq}}$ and $F(S_{\text{non-eq}})$ as inputs and $\text{Noise}(S_{\text{non-eq}}, \tilde{S}_{\text{non-eq}})$ as outputs, they implicitly learn to leverage $F(S_{\text{non-eq}})$ to reconstruct $S_{\text{non-eq}}$ instead of predicting any arbitrary non-equilibrium structures. Empirically, we find that force encoding is critical to the effectiveness of DeNS on OC20 S2EF-2M dataset (Index 2 and Index 3 in Table 1(b)) and MD17 dataset (Index 2 and Index 3 in Table 6).

Since DeNS requires encoding forces, DeNS favors equivariant networks, which can easily incorporate forces as well as other higher-degree tensors like stress into their node embeddings. Specifically, the node embeddings of equivariant networks are equivariant irreps features built from vectors spaces of irreducible representations (irreps) and contain $C_L$ channels of type-$L$ vectors with degree $L$ being in the range from 0 to maximum degree $L_{max}$. $C_L$ and $L_{max}$ are architectural hyper-parameters of equivariant networks. To obtain the force embedding $x_{\mathbf{f}}$ from the input force $\mathbf{f}$, we first project $\mathbf{f}$ into type-$L$ vectors with spherical harmonics $Y^{(L)}\left(\frac{\mathbf{f}}{||\mathbf{f}||}\right)$, where $0 \leqslant L \leqslant L_{max}$, and then expand the number of channels from 1 to $C_L$ with an $SO(3)$ linear layer (Geiger et al., 2022; Geiger & Smidt, 2022; Liao & Smidt, 2023):

$$x_{\mathbf{f}}^{(L)} = \text{SO3\_Linear}^{(L)}\left( ||\mathbf{f}|| \cdot Y^{(L)}\left(\frac{\mathbf{f}}{||\mathbf{f}||}\right) \right) \tag{5}$$

$x_{\mathbf{f}}^{(L)}$ denotes the channels of type-$L$ vectors in force embedding $x_{\mathbf{f}}$, and $\text{SO3\_Linear}^{(L)}$ denotes the $SO(3)$ linear operation on type-$L$ vectors. Since we normalize the input force when using spherical harmonics, we multiply $Y^{(L)}\left(\frac{\mathbf{f}}{||\mathbf{f}||}\right)$ with the norm of input force $||\mathbf{f}||$ to recover the information of force magnitude. After computing force embeddings for all atom-wise forces, we simply add the force embeddings to initial node embeddings to encode forces in equivariant networks. The pseudocode for encoding forces into node embeddings can be found in Section E.

On the other hand, it might not be that intuitive to encode forces in invariant networks since their internal latent representations such as node embeddings and edge embeddings are scalars instead of type-$L$ vectors

or geometric tensors. One potential manner of encoding forces in latent representations is to project them into edge embeddings by taking inner products between forces and edge vectors of relative positions. This process is the inverse of how GemNet-OC (Gasteiger et al., 2022) decodes forces from latent representations. Since equivariant networks have been shown to outperform invariant networks on atomistic datasets and are a more natural fit to encoding forces, we mainly focus on equivariant networks and how DeNS can further advance their performance. As for other work (Duval et al., 2023) leveraging frame averaging to achieve equivariance through data transformations, we can follow how unit cell Cartesian coordinates are projected in their framework to encode forces when optimizing for DeNS. Specifically, we compute one set of frame axes using only 3D atomic positions and project the input forces to the frame axes. The projected forces remain the same under any $E(3)$ transformation and enable using unconstrained functions to encode the input forces.

### 3.2.3 Training with DeNS

**Auxiliary Task.** We propose to use DeNS as an auxiliary task along with the original task of predicting energy and forces and summarize the training process in Figure 2. Specifically, given a batch of structures, for each structure, we decide whether we optimize the objective of DeNS (Figure 2(b) or Figure 2(c)) or the objective of the original task (Figure 2(a)). This introduces an additional hyper-parameter $p_{\text{DeNS}}$, the probability of optimizing DeNS. We use an additional noise head for noise prediction, which slightly increases training time. Additionally, when training DeNS, similar to Noisy Nodes (Godwin et al., 2022), we also leverage energy labels and predict the energy of original structures. Therefore, given an original non-equilibrium structure $S_{\text{non-eq}}$ and the corrupted counterpart $\tilde{S}_{\text{non-eq}}$, training DeNS corresponds to minimizing the following loss function:

$$\lambda_E \cdot \mathcal{L}_E + \lambda_{\text{DeNS}} \cdot \mathcal{L}_{\text{DeNS}} = \lambda_E \cdot \left| E'(S_{\text{non-eq}}) - \hat{E}\left(\tilde{S}_{\text{non-eq}}, F(S_{\text{non-eq}})\right)\right| + \lambda_{\text{DeNS}} \cdot \mathcal{L}_{\text{DeNS}} \tag{6}$$

where $\mathcal{L}_{\text{DeNS}}$ denotes the loss function of denoising as defined in Equation 4. We note that we also encode forces as discussed in Section 3.2.2 to predict the energy of $S_{\text{non-eq}}$, and we share the energy prediction head across Equation 1 and Equation 6. The loss function introduces another hyper-parameter $\lambda_{\text{DeNS}}$, DeNS coefficient, controlling the relative importance of the auxiliary task. Besides, the process of corrupting structures also results in another hyper-parameter $\sigma$ as shown in Equation 2. We provide the pseudocode in Section F. We note that the force encoding in DeNS only relies on force labels on the training set and we do not use any force labels on the validation or testing sets.

**Partially Corrupted Structures.** Empirically, we find that adding noise to all atoms in a structure can sometimes lead to limited performance gain of DeNS. We surmise adding noise to all atoms makes denoising too difficult and potentially not well-defined and that there are still several structures satisfying input forces. To address this issue, we use partially corrupted structures, where we only add noise to and denoise a random subset of atoms as shown in Figure 2(c). Specifically, for corrupted atoms, we encode their atom-wise forces and predict the noise. For other uncorrupted atoms, we do not encode forces and train for the original task of predicting forces. We also predict the energy of the original structures given partially corrupted structures. This introduces an additional hyper-parameter, corruption ratio $r_{\text{DeNS}}$, which is the ratio of the number of corrupted atoms to that of all atoms.

### 3.2.4 Discussion

**Why Does DeNS Help.** DeNS can better leverage training data to improve the performance in the following two ways. First, DeNS adds noise to non-equilibrium structures to generate structures with new geometries and therefore naturally achieves data augmentation (Godwin et al., 2022). Second, training DeNS enourages learning a different yet highly correlated interaction. Since we encode forces as inputs and predict the original structures in terms of noise corrections, DeNS enables learning the interaction of transforming forces into structures, which is the inverse of predicting forces. As demonstrated in the works of Noisy Nodes (Godwin et al., 2022) and UL2 (Tay et al., 2023), training a single model with multiple correlated objectives to learn different interactions can help the performance on original tasks.

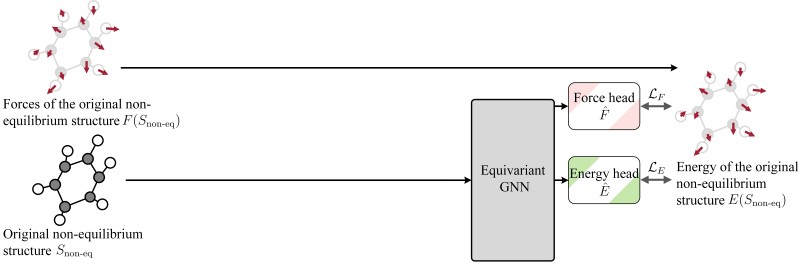

(a) Original task of predicting energy and forces.

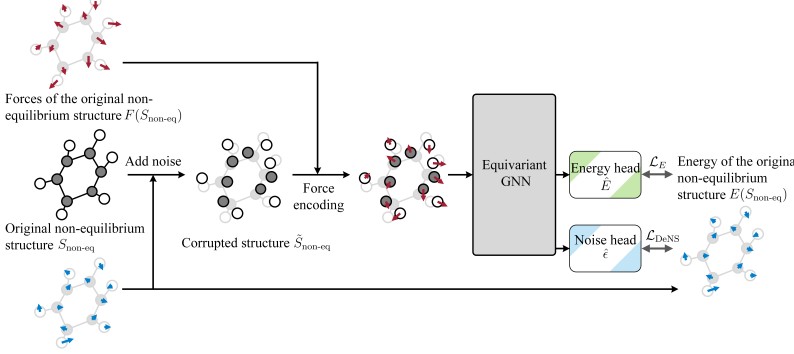

(b) DeNS without partially corrupted structures. All the atoms in a structure are corrupted with Gaussian noise. We encode all the atom-wise forces to predict noise.

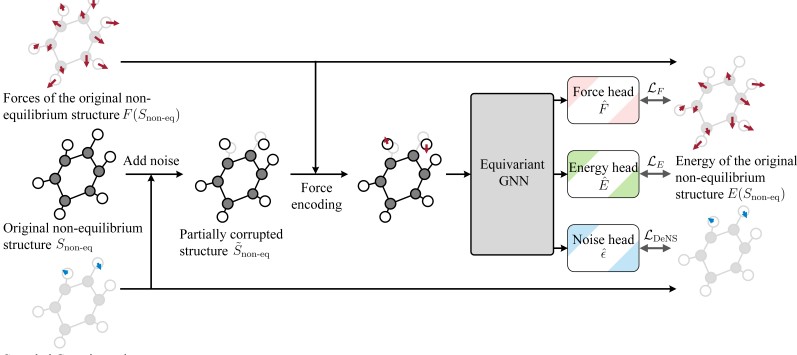

(c) DeNS with partially corrupted structures. We add noise to and denoise a random subset of atoms. For corrupted atoms, we encode their forces and predict the noise. For other uncorrupted atoms, we instead predict their forces. Here only the top two white atoms are corrupted.

Figure 2: Training process when incorporating DeNS as an auxiliary task. The pseudocode for training with DeNS can be found in Section F. "Equivariant GNN", "energy head", "force head" and "noise head" are shared across (a), (b) and (c). For each batch of structures, we use the original task (a) for some structures and DeNS ((b) or (c)) for the others. Using partially corrupted structures as in (c) is empirically better than (b). We note that the force label $F(S_{\text{non-eq}})$ and energy label $E(S_{\text{non-eq}})$ used in (b) and (c) are the same as those in (a) and that training with DeNS does not require any additional data.

**Connection to Self-Supervised Learning.** DeNS shares similar intuitions to self-supervised learning methods like BERT (Devlin et al., 2019) and MAE (He et al., 2022) and other denoising methods (Vincent et al., 2008; 2010; Godwin et al., 2022; Zaidi et al., 2023) – they remove or corrupt a portion of input data and then learn to predict the original data. Learning to reconstruct data can help learning generalizable representations, and therefore these methods can use the task of reconstruction to improve the performance on downstream tasks. However, unlike those self-supervised learning methods (Devlin et al., 2019; He et al., 2022; Zaidi et al., 2023), DeNS requires force labels for encoding, and therefore, we propose to use DeNS as an

**(a) Comparison of training with and without DeNS.**

| | EquiformerV2 | | | | EquiformerV2 + DeNS | | | |
|---|---|---|---|---|---|---|---|---|
| Epochs | forces | energy | Number of parameters | training time (GPU-hours) | forces | energy | Number of parameters | training time (GPU-hours) |
| 12 | 20.46 | 285 | 83M | 1398 | 19.09 | 269 | 89M | 1501 |
| 20 | 19.78 | 280 | 83M | 2330 | 18.58 | 260 | 89M | 2501 |
| 30 | 19.42 | 278 | 83M | 3495 | 18.02 | 251 | 89M | 3752 |
| | eSCN | | | | eSCN + DeNS | | | |
| 20 | 20.61 | 290 | 52M | 1802 | 19.14 | 268 | 52M | 1829 |

**(b) Design Choices.**

| | Method | | | | | |
|---|---|---|---|---|---|---|
| Index | DeNS | Force encoding | $\lambda_E \neq 0$ in Eq. 6 | Partially corrupted structures | forces | energy |
| 1 | | | | | 20.46 | 285 |
| 2 | ✓ | ✓ | ✓ | ✓ | 19.09 | 269 |
| 3 | ✓ | | ✓ | ✓ | 21.32 | 278 |
| 4 | ✓ | ✓ | ✓ | | 18.87 | 285 |
| 5 | ✓ | ✓ | ✓ | | 19.54 | 279 |

Table 1: Ablation results of training with DeNS on OC20 S2EF-2M dataset. We report mean absolute errors for forces in meV/Å and energy in meV, and lower is better. Errors are averaged over the four validation sub-splits of OC20. (a) We train EquiformerV2 and eSCN and compare the results of training with and without DeNS. The training time is measured on V100 GPUs. (b) We train EquiformerV2 for 12 epochs to verify the design choices of DeNS.

auxiliary task optimized along with original tasks and do not follow the previous practice of first pre-training and then fine-tuning. Additionally, we note that before obtaining a single equilibrium structure, we need to run relaxations and generate many intermediate non-equilibrium ones (Figure 1(a)), which is the labeling process as well. We hope that the ability to leverage more from non-equilibrium structures as proposed in this work can encourage researchers to release data containing intermediate non-equilibrium structures in addition to final equilibrium ones. Moreover, we note that DeNS can also be used in fine-tuning. For example, we can first pre-train models on PCQM4Mv2 dataset and then fine-tune them on the smaller MD17 dataset with both the original task and DeNS.

**Marginal Increase in Training Time.** Since we use an additional noise head for denoising, training with DeNS marginally increases the time of each training iteration. We optimize DeNS for some structures and the original task for the others for each training iteration, and we demonstrate that DeNS can improve the performance given the same amount of training iterations. Therefore, training with DeNS only marginally increase the overall training time.

## 4 Experiments

### 4.1 OC20 Dataset

**Dataset and Tasks.** We start with experiments on the large and diverse Open Catalyst 2020 dataset (OC20) (Chanussot* et al., 2021), which consists of about 1.2M Density Functional Theory (DFT) relaxation trajectories. Each DFT trajectory in OC20 starts from an initial structure of an adsorbate molecule placed on a catalyst surface, which is then relaxed with the revised Perdew-Burke-Ernzerhof (RPBE) functional (Hammer et al., 1999) calculations to a local energy minimum. Relevant to DeNS, all the intermediate structures from a relaxation trajectory, except the relaxed structure, are considered non-equilibrium structures. The relaxed or equilibrium structure has forces close to zero.

The primary task in OC20 is Structure to Energy and Forces (S2EF), which is to predict the energy and per-atom forces given an equilibrium or non-equilibrium structure from any point in the trajectory. These predictions are evaluated on energy and force mean absolute error (MAE). Most of the previous works use direct methods to predict forces on OC20, and we follow this practice and investigate how DeNS can improve direct methods for force prediction. Once a model is trained for S2EF, it is used to run structural relaxations from an initial structure using the predicted forces till a local energy minimum is found. The energy predictions of these relaxed structures are evaluated on the Initial Structure to Relaxed Energy (IS2RE) task.

**Training Details.** Please refer to Section B.1 for details on DeNS, architectures, hyper-parameters and training time.

### 4.1.1 Ablation Studies

We use OC20 S2EF-2M dataset to compare the results of training with and without DeNS and verify the design choices of DeNS.

**Comparison between Training with and without DeNS.**    Table 1(a) summarizes the results of training with and without DeNS. For EquiformerV2, incorporating DeNS as an auxiliary task boosts the performance on energy and force predictions and only increases training time by 7.4% and the number of parameters from 83M to 89M. Particularly, EquiformerV2 trained with DeNS for 12 epochs outperforms EquiformerV2 trained without DeNS for 30 epochs and saves 2.3× training time. This suggests that using data augmentation and learning an auxiliary task are more efficient to improve performance compared to simply training for longer. Additionally, we show that DeNS can be applicable to other equivariant networks like eSCN (Passaro & Zitnick, 2023). Training eSCN with DeNS for 20 epochs results in better energy and force MAE compared to EquiformerV2 trained without DeNS for 30 epochs while requiring 1.9× less training time. DeNS slightly increases training time of eSCN by 1.5%, and the different amounts of increase in training time between EquiformerV2 and eSCN are because they use different modules for noise prediction.

**Design Choices.**    We train EquiformerV2 for 12 epochs with DeNS to verify the design choices of DeNS and summarize the results in Table 1(b). Comparing Index 2 and Index 3, we show that encoding forces $F(S_{\text{non-eq}})$ in Equations 4 and 6 enables denoising non-equilibrium structures to significantly improve both energy and force MAE. Moreover, DeNS without force encoding (Index 3) results in clearly worse force MAE compared to training without DeNS (Index 1). These results verify our claim that denoising non-equilibrium structures naively can be ill-posed and potentially harmful to performance and that force encoding is critical to leverage the gain of denoising. We note that force encoding only relies on force labels in the training set and does not require any additional data. We also compare DeNS with and without force encoding on MD17 in Section 4.3 and have similar observations. Comparing Index 2 and Index 4, we demonstrate that predicting energy of original structures given corrupted ones can improve energy MAE by 5.6% while slightly increasing force MAE by 1.2%. The increase in force MAE is because predicting energy given corrupted structures is equivalent to using $\lambda_E \neq 0$ in Equation 6 and implicitly decreases the relative importance of force prediction. Since the decrease in energy MAE is greater than the increase in force MAE, we adopt the practice of predicting energy given corrupted structures. Finally, the comparison between Index 2 and Index 5 shows that partially corrupted structures can further improve the performance gain of DeNS.

### 4.1.2 Main Results

We train EquiformerV2 (160M) with DeNS on OC20 S2EF-All+MD dataset. The model follows the same configuration as EquiformerV2 (153M) trained without DeNS, and the additional parameters are due to force encoding and one additional block of equivariant graph attention for noise prediction. We report results in Table 2. All test results are computed via the EvalAI evaluation server[1]. EquiformerV2 trained with DeNS achieves new state-of-the-art results on both S2EF and IS2RE tasks. On the S2EF validation split, EquiformerV2 trained with DeNS improves energy MAE by 5meV and force MAE by 1.0meV/Å, which is comparable to the gain brought by increasing the size of EquiformerV2 from 31M to 153M (energy MAE improved by 5meV and force MAE improved by 1.3meV/Å). On the S2EF test split, the improvement in energy and force predictions is smaller, which is probably because of different splits. On the IS2RE test split, training EquiformerV2 with DeNS reduces MAE by 16meV, achieving similar performance gain of going from eSCN (Passaro & Zitnick, 2023) to EquiformerV2 (Liao et al., 2023) (IS2RE MAE improved by 14meV) and demonstrating the effectiveness of training with DeNS. We note that the improvement might not be as significant as that on OC20 S2EF-2M (Section 4.1.1), OC22 (Section 4.2) and MD17 (Section 4.3) datasets since the OC20 S2EF-All+MD training set contains much more structures along relaxation trajectories, making new 3D geometries generated by DeNS less helpful. However, DeNS is still valuable because most datasets are not as large as OC20 S2EF-All+MD dataset (about 172M structures in the training set) but have sizes closer to OC20 S2EF-2M (2M structures), OC22 (8.2M structures), and MD17 (950 structures) datasets.

---

[1]eval.ai/web/challenges/challenge-page/712

| Model | Throughput | S2EF validation | | S2EF test | | IS2RE test |
|---|---|---|---|---|---|---|
| | Samples / GPU sec. ↑ | Energy MAE (meV) ↓ | Force MAE (meV/Å) ↓ | Energy MAE (meV) ↓ | Force MAE (meV/Å) ↓ | Energy MAE (meV) ↓ |
| GemNet-OC-L-E (Gasteiger et al., 2022) | 7.5 | 239 | 22.1 | 230 | 21.0 | - |
| GemNet-OC-L-F (Gasteiger et al., 2022) | 3.2 | 252 | 20.0 | 241 | 19.0 | - |
| GemNet-OC-L-F+E (Gasteiger et al., 2022) | - | - | - | - | - | 348 |
| SCN L=6 K=16 (4-tap 2-band) (Zitnick et al., 2022) | - | - | - | 228 | 17.8 | 328 |
| SCN L=8 K=20 (Zitnick et al., 2022) | - | - | - | 237 | 17.2 | 321 |
| eSCN L=6 K=20 (Passaro & Zitnick, 2023) | 2.9 | 243 | 17.1 | 228 | 15.6 | 323 |
| EquiformerV2 ($\lambda_E = 4$, 31M) (Liao et al., 2023) | 7.1 | 232 | 16.3 | 228 | 15.5 | 315 |
| EquiformerV2 ($\lambda_E = 4$, 153M) (Liao et al., 2023) | 1.8 | 227 | 15.0 | 219 | 14.2 | 309 |
| EquiformerV2 + DeNS ($\lambda_E = 4$, 160M) | 1.8 | **222** | **14.0** | **214** | **13.3** | **293** |

Table 2: OC20 results on S2EF validation and test splits and IS2RE test split when trained on OC20 S2EF-All+MD dataset. Throughput is reported as the number of structures processed per GPU-second during training and measured on V100 GPUs.

| Model | Number of parameters | S2EF-Total validation | | | | S2EF-Total test | | | | IS2RE-Total test | |
|---|---|---|---|---|---|---|---|---|---|---|---|
| | | Energy MAE (meV) ↓ | | Force MAE (meV/Å) ↓ | | Energy MAE (meV) ↓ | | Force MAE (meV/Å) ↓ | | Energy MAE (meV) ↓ | |
| | | ID | OOD | ID | OOD | ID | OOD | ID | OOD | ID | OOD |
| GemNet-OC (Gasteiger et al., 2022) | 39M | 545 | 1011 | 30 | 40 | 374 | 829 | 29.4 | 39.6 | 1329 | 1584 |
| GemNet-OC (trained on OC20 + OC22) (Gasteiger et al., 2022) | 39M | 464 | 859 | 27 | 34 | 311 | 689 | 26.9 | 34.2 | 1200 | 1534 |
| eSCN (Passaro & Zitnick, 2023) | 200M | - | - | - | - | 350 | 789 | 23.8 | 35.7 | - | - |
| EquiformerV2 ($\lambda_E = 4, \lambda_F = 100$) (Liao et al., 2023) | 122M | 433 | 629 | 22.88 | 30.70 | 263.7 | 659.8 | 21.58 | 32.65 | 1119 | 1440 |
| EquiformerV2 + DeNS ($\lambda_E = 4, \lambda_F = 100$) | 127M | **391.6** | **533.0** | **20.66** | **27.11** | **236.4** | **590.7** | **20.04** | **29.31** | **951** | **1282** |

Table 3: OC22 results on S2EF-Total validation and test splits and IS2RE-Total test split. Models are trained on the OC22 S2EF-Total training split unless otherwise noted.

## 4.2 OC22 Dataset

**Dataset and Tasks.** The Open Catalyst 2022 (OC22) dataset (Tran* et al., 2022) focuses on oxide electrocatalysis and consists of about 62k DFT relaxations obtained with Perdew-Burke-Ernzerhof (PBE) functional calculations. One crucial difference between OC22 and OC20 is that the energy targets in OC22 are DFT total energies. DFT total energies are harder to predict but are the most general and closest to a DFT surrogate, offering the flexibility to study property prediction beyond adsorption energies. Analogous to the task definitions in OC20, the primary tasks in OC22 are S2EF-Total and IS2RE-Total. We train models on the OC22 S2EF-Total dataset, which has 8.2M structures, and evaluate them on energy and force MAE on the S2EF-total validation and test splits. Same as OC20, we use direct methods to predict forces here. Then, we use these models to perform relaxations starting from initial structures in the IS2RE-Total test split and evaluate the predicted relaxed energies on energy MAE.

**Training Details.** Please refer to Section C.1 for details on architectures, hyper-parameters and training time.

**Results.** We use EquiformerV2 with energy coefficient $\lambda_E = 4$ and force coefficient $\lambda_F = 100$ to demonstrate how DeNS can further improve state-of-the-art results and summarize the comparison in Table 3. Compared to EquiformerV2 ($\lambda_E = 4$, $\lambda_F = 100$) trained without DeNS, EquiformerV2 trained with DeNS consistently achieves better energy and force MAE across all the S2EF-Total validation and test splits, with 9.6% to 15.3% improvement in energy MAE and 7.1% to 11.7% improvement in force MAE. For IS2RE-Total, EquiformerV2 trained with DeNS achieves the best energy MAE results. The improvement in IS2RE-Total from training with DeNS on only OC22 is greater than that of training on the much larger OC20 and OC22 datasets reported in previous works. Specifically, training GemNet-OC on OC20 and OC22 datasets (about 134M + 8.2M structures) improves IS2RE-Total energy MAE ID by 129meV and OOD by 50meV compared to training GemNet-OC on only OC22 dataset (8.2M structures). Compared to training without DeNS, training EquiformerV2 with DeNS improves ID by 168meV and OOD by 158meV. Thus, training with DeNS clearly improves data efficiency and performance on OC22.

| Model | Aspirin energy | Aspirin forces | Benzene energy | Benzene forces | Ethanol energy | Ethanol forces | Malonaldehyde energy | Malonaldehyde forces | Naphthalene energy | Naphthalene forces | Salicylic acid energy | Salicylic acid forces | Toluene energy | Toluene forces | Uracil energy | Uracil forces | Training time (GPU-hours)↓ | Number of parameters |
|---|---|---|---|---|---|---|---|---|---|---|---|---|---|---|---|---|---|---|
| SchNet (Schütt et al., 2017) | 16.0 | 58.5 | 3.5 | 13.4 | 3.5 | 16.9 | 5.6 | 28.6 | 6.9 | 25.2 | 8.7 | 36.9 | 5.2 | 24.7 | 6.1 | 24.3 | - | - |
| DimeNet (Gasteiger et al., 2020) | 8.8 | 21.6 | 3.4 | 8.1 | 2.8 | 10.0 | 4.5 | 16.6 | 5.3 | 9.3 | 5.8 | 16.2 | 4.4 | 9.4 | 5.0 | 13.1 | - | - |
| PaiNN (Schütt et al., 2021) | 6.9 | 14.7 | - | - | 2.7 | 9.7 | 3.9 | 13.8 | 5.0 | 3.3 | 4.9 | 8.5 | 4.1 | 4.1 | 4.5 | 6.0 | - | - |
| TorchMD-NET (Thölke & Fabritiis, 2022) | 5.3 | 11.0 | 2.5 | 8.5 | 2.3 | 4.7 | 3.3 | 7.3 | **3.7** | 2.6 | **4.0** | 5.6 | **3.2** | 2.9 | **4.1** | 4.1 | - | - |
| NequIP ($L_{max}=3$) (Batzner et al., 2022) | 5.7 | 8.0 | - | - | **2.2** | 3.1 | 3.3 | 5.6 | 4.9 | 1.7 | 4.6 | 3.9 | 4.0 | 2.0 | 4.5 | 3.3 | - | - |
| Equiformer ($L_{max}=2$) | 5.3 | 7.2 | **2.2** | 6.6 | **2.2** | 3.1 | 3.3 | 5.8 | **3.7** | 2.1 | 4.5 | 4.1 | 3.8 | 2.1 | 4.3 | 3.3 | 17 | 3.50M |
| Equiformer ($L_{max}=3$) | 5.3 | 6.6 | 2.5 | 8.1 | **2.2** | 2.9 | **3.2** | 5.4 | 4.4 | 2.0 | 4.3 | 3.9 | 3.7 | 2.1 | 4.3 | 3.4 | 59 | 5.50M |
| Equiformer ($L_{max}=2$) + DeNS | 5.1 | 5.7 | 2.3 | **6.1** | **2.2** | 2.6 | **3.2** | 4.4 | **3.7** | 1.7 | 4.4 | 3.7 | 3.5 | 1.9 | 4.2 | 3.3 | 19 | 4.00M |
| Equiformer ($L_{max}=3$) + DeNS | **5.0** | **5.2** | 2.3 | **6.1** | **2.2** | 2.4 | **3.2** | **4.1** | **3.7** | **1.6** | 4.2 | **3.2** | 3.4 | **1.8** | **4.1** | **2.9** | 61 | 6.35M |

Table 4: Mean absolute error results on the MD17 testing set. Energy and force are in units of meV and meV/Å. We additionally report the time of training different Equiformer models averaged over all molecules and the numbers of parameters to show that the proposed DeNS can improve performance with minimal overhead.

| Index | Attention | Layer normalization | DeNS | Aspirin energy | Aspirin forces | Benzene energy | Benzene forces | Ethanol energy | Ethanol forces | Malonaldehyde energy | Malonaldehyde forces | Naphthalene energy | Naphthalene forces | Salicylic acid energy | Salicylic acid forces | Toluene energy | Toluene forces | Uracil energy | Uracil forces |
|---|---|---|---|---|---|---|---|---|---|---|---|---|---|---|---|---|---|---|---|
| 1 | ✓ | ✓ | | 5.3 | 7.2 | **2.2** | 6.6 | **2.2** | 3.1 | 3.3 | 5.8 | **3.7** | 2.1 | 4.5 | 4.1 | 3.8 | 2.1 | 4.3 | **3.3** |
| 2 | ✓ | ✓ | ✓ | **5.1** | **5.7** | 2.3 | **6.1** | **2.2** | **2.6** | **3.2** | **4.4** | **3.7** | **1.7** | 4.4 | **3.7** | 3.5 | **1.9** | 4.2 | 3.3 |
| 3 | | ✓ | | 5.2 | 7.7 | 2.4 | 6.2 | 2.3 | 3.9 | 3.3 | 6.2 | 3.8 | 2.2 | **4.1** | 4.7 | **3.3** | 2.4 | **4.2** | 4.4 |
| 4 | | ✓ | ✓ | 5.2 | **6.1** | 2.4 | **6.1** | **2.2** | 2.9 | **3.2** | 5.1 | **3.7** | **1.7** | 4.2 | 3.9 | 3.4 | 2.0 | **4.2** | 3.4 |
| 5 | | | | 5.3 | 9.3 | 2.4 | 9.2 | 2.3 | 4.5 | 3.4 | 8.2 | **3.7** | 2.4 | 4.2 | 5.5 | **3.3** | 2.9 | **4.2** | 4.8 |
| 6 | | | ✓ | 5.2 | 7.3 | 2.4 | 8.1 | **2.2** | 3.4 | 3.4 | 6.7 | **3.7** | 1.9 | 4.2 | 4.4 | 3.4 | 2.2 | **4.2** | 3.8 |

Table 5: Mean absolute error results of variants of Equiformer ($L_{max}=2$) without attention and layer normalization on the MD17 testing set. Energy and force are in units of meV and meV/Å. Index 1 and Index 2 correspond to "Equiformer ($L_{max}=2$)" and "Equiformer ($L_{max}=2$) + DeNS" in Table 4.

| Index | DeNS | Force encoding | $\lambda_E \neq 0$ in Eq. 6 | Partially corrupted structures | Aspirin energy | Aspirin forces | Benzene energy | Benzene forces | Ethanol energy | Ethanol forces | Malonaldehyde energy | Malonaldehyde forces | Naphthalene energy | Naphthalene forces | Salicylic acid energy | Salicylic acid forces | Toluene energy | Toluene forces | Uracil energy | Uracil forces |
|---|---|---|---|---|---|---|---|---|---|---|---|---|---|---|---|---|---|---|---|---|
| 1 | | | | | 5.3 | 7.2 | **2.2** | 6.6 | **2.2** | 3.1 | 3.3 | 5.8 | **3.7** | 2.1 | 4.5 | 4.1 | 3.8 | 2.1 | 4.3 | **3.3** |
| 2 | ✓ | ✓ | ✓ | ✓ | **5.1** | **5.7** | 2.3 | **6.1** | **2.2** | **2.6** | **3.2** | **4.4** | **3.7** | **1.7** | 4.4 | 3.7 | **3.5** | **1.9** | 4.2 | **3.3** |
| 3 | ✓ | | ✓ | ✓ | 8.6 | 9.1 | 2.3 | 6.3 | 2.3 | 3.3 | **3.2** | 5.8 | 7.7 | 6.1 | 5.2 | 10.6 | 3.7 | 2.0 | 5.5 | 6.5 |
| 4 | ✓ | ✓ | | ✓ | **5.1** | 5.8 | 2.3 | **6.1** | 2.3 | **2.6** | **3.2** | 4.5 | **3.7** | **1.7** | 4.8 | **3.6** | 3.7 | **1.9** | 4.2 | **3.3** |
| 5 | ✓ | ✓ | ✓ | | 5.3 | 6.9 | 2.4 | 6.3 | 2.3 | 3.2 | 3.3 | 5.4 | 3.7 | 2.0 | 4.8 | 4.2 | 4.0 | 2.0 | **4.2** | 3.8 |

Table 6: Ablation study on the design choices of DeNS using MD17 dataset. Mean absolute error results are evaluated on the testing set. Energy and force are in units of meV and meV/Å. Index 1 and Index 2 correspond to "Equiformer ($L_{max}=2$)" and "Equiformer ($L_{max}=2$) + DeNS" in Table 4.

## 4.3 MD17 Dataset

**Dataset.** The MD17 dataset (Chmiela et al., 2017; Schütt et al., 2017; Chmiela et al., 2018) consists of molecular dynamics simulations of small organic molecules. The task is to predict the energy and forces of these non-equilibrium molecules. Following previous works, we adopt gradient methods for force prediction. We use 950 and 50 different configurations for training and validation sets and the rest for the testing set.

**Training Details.** Please refer to Section D.2 for additinoal implementation details of DeNS, hyperparameters and training time.

**Main Results.** We train Equiformer ($L_{max} = 2$) (Liao & Smidt, 2023) and Equiformer ($L_{max} = 3$) with DeNS based on their official implementation, where $L_{max}$ denotes the maximum degree of equivariant representations. As shown in Table 4, DeNS improves the results on all molecules. Along with the results on OC20 and OC22 datasets, DeNS can generally improve the performance on force predictions with both direct (i.e., OC20 and OC22) and gradient (i.e., MD17) methods. Particularly, Equiformer ($L_{max} = 2$) trained with DeNS acheives better results on all the tasks and requires $3.1\times$ less training time than Equiformer ($L_{max} = 3$) trained without DeNS. This demonstrates that for this small dataset, training an auxiliary task and using data augmentation are more efficient and result in larger performance gain than increasing $L_{max}$ from 2 to 3. Additionally, we find that the gains from training DeNS as an auxiliary task are comparable to pre-training. For example, Zaidi et al. (2023) uses TorchMD-NET (Thölke & Fabritiis, 2022) pre-trained on the PCQM4Mv2 dataset and reports results on Aspirin. Their improvement in force MAE is about 17.2% (Table 3 in Zaidi et al. (2023)). Training Equiformer ($L_{max} = 2$) with DeNS results in 20.8% improvement in force MAE without relying on another dataset. Note that we only increase training time by 10.5% while their method takes much more time since PCQM4Mv2 dataset is more than 3000× larger than the training set of

MD17. Moreover, training with DeNS enables consistent improvement in all molecules when increasing $L_{max}$ from 2 to 3, and Equiformer ($L_{max} = 3$) trained with DeNS achieves overall best results. In contrast, when DeNS is not used, increasing $L_{max}$ from 2 to 3 can lead to overfitting and worse results on some molecules (i.e., Benzene and Uracil). Besides, we report simulation-based results in Section D.3.

**Effect of DeNS on Different Network Architectures.** We train variants of Equiformer ($L_{max} = 2$) by removing attention and layer normalization to investigate the performance gain of DeNS on different network archtiectures. The results are summarized in Table 5, and DeNS improves all the model variants. We note that Equiformer without attention and layer normalization reduces to SEGNN (Brandstetter et al., 2022) but with a better radial basis function. Since the models cover many variants of equivariant networks, this suggests that DeNS is general and can be helpful to many equivariant networks.

**Ablation Study on the Design Choices of DeNS.** We use Equiformer ($L_{max} = 2$) to justify the design choices of DeNS. The results are summarized in Table 6 and are similar to those on OC20 S2EF-2M dataset. Comparing Index 2 and Index 3, force encoding consistently results in significant improvement in energy and force MAE. Compared to training without denoising (Index 1), DeNS without force encoding (Index 3) only achieves slightly better results on some molecules (i.e., Benzene, Malondaldehyde, and Toluene) and much worse results on others. For molecules on which DeNS without force encoding is helpful, adding force encoding can achieve even better results. For others, force encoding is indispensable for DeNS to be effective. Comparing Index 2 and Index 4, predicting energy given corrupted structures results in overall better performance. Finally, the comparison between Index 2 and Index 5 demonstrates that partially corrupted structures are necessary to achieve better results. For some molecules (i.e., Ethanol and Salicycli acid), DeNS with noise added to all atoms (Index 5) can be worse than training without DeNS (Index 1).

## 5  Conclusion

In this paper, we propose to use **de**noising **n**on-equilibrium **s**tructures (DeNS) as an auxiliary task to better leverage training data and improve performance on original tasks of energy and force predictions. Denoising non-equilibrium structures can be an ill-posed problem since there are many possible target structures. To address the issue, we propose force encoding and take the forces of original structures as inputs to specify which non-equilibrium structures we are denoising. With force encoding, DeNS successfully improves the performance on original tasks when it is used as an auxiliary task. We conduct extensive experiments on OC20, OC22 and MD17 datasets to demonstrate that DeNS can boost the performance on energy and force predictions across datasets of various scales with minimal increase in training cost and is applicable to many equivariant networks. Finally, we note that the proposed DeNS is general and can be directly applied to other atomistic datasets containing non-equilibrium structures. Take Materials Project (Jain et al., 2013) for example. Similar to OC20 and OC22 datasets, for each entry in the Materials Project database, the Materials Project Trajectory (MPtrj) dataset (Deng et al., 2023) contains the corresponding relaxation trajectory between initial and relaxed structures. Most structures in MPtrj are non-equilibrium and have energy and force labels. Therefore, we can apply DeNS to MPtrj dataset in the same way as OC20 and OC22 datasets.

## Acknowledgement

We thank Larry Zitnick, Xinlei Chen, Zachary Ulissi, Saro Passaro, Anuroop Sriram, and Brandon Wood for helpful discussions. We acknowledge the MIT SuperCloud and Lincoln Laboratory Supercomputing Center (Reuther et al., 2018) for providing high performance computing and consultation resources that have contributed to the research results reported within this paper.

Yi-Lun Liao and Tess Smidt were supported by DOE ICDI grant DE-SC0022215.

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

## Appendix

## A   Related Works

### A.1   Comparison to Previous Works on Denoising

We discuss previous works on denoising (Godwin et al., 2022; Zaidi et al., 2023; Feng et al., 2023b; Wang et al., 2023) in chronological order and compare them with this work as below.

Godwin et al. (2022) first proposes the idea of adding noise to 3D coordinates and then using denoising as an auxiliary task. The auxiliary task is trained along with the original task without relying on another large dataset. Their approach requires known equilibrium structures and therefore is limited to QM9 (Ramakrishnan et al., 2014; Ruddigkeit et al., 2012) and OC20 IS2RE datasets and can not be applied to force prediction such as OC20 S2EF dataset. For QM9, all the structures are at equilibrium, and for OC20 IS2RE, the target of denoising is the relaxed, equilibrium structure. Denoising without force encoding is well-defined on both QM9 and OC20 IS2RE datasets. In contrast, this work proposes using force encoding to generalize their approach to non-equilibrium structures, which have much larger datasets than equilibrium ones. Force encoding can achieve better results on OC20 S2EF-2M dataset with little overhead (Index 2 and Index 3 in Table 1(b)) and is indispensable on MD17 dataset (Section 4.3).

Zaidi et al. (2023) adopts the denoising approach proposed by Godwin et al. (2022) as a pre-training method and therefore requires another large dataset containing unlabelled equilibrium structures for pre-training. On the other hand, Godwin et al. (2022) and this work use denoising along with the original task and do not use any additional unlabeled data.

Feng et al. (2023b) follows the same practice of pre-training via denoising (Zaidi et al., 2023) and proposes a different manner of adding noise. Specifically, they separate noise into dihedral angle noise and coordinate noise and only learn to predict coordinate noise. However, adding noise to dihedral angles requires tools like RDKit (rdk) to obtain rotatable bonds and cannot be applied to other datasets like OC20 and OC22.

Although Zaidi et al. (2023) and Feng et al. (2023b) report results of force prediction on MD17 dataset, they first pre-train models on PCQM4Mv2 dataset (Nakata & Shimazaki, 2017) and then fine-tune the pre-trained models on MD17 dataset. We note that their setting is different from ours since we do not use any dataset for pre-training. As for fine-tuning on MD17 dataset, Zaidi et al. (2023) simply follows the same practice

in standard supervised training. Feng et al. (2023b) explores fine-tuning with objectives similar to Noisy Nodes (Godwin et al., 2022), but the performance gain is much smaller than ours. Concretely, in Table 5 in Feng et al. (2023b), the improvement in force prediction on Aspirin is about 2.6% while we improve force MAE by 20.8%.

Wang et al. (2023) uses the same pre-training method as Zaidi et al. (2023) but applies it to ANI-1 (Smith et al., 2017) and ANI-1x (Smith et al., 2018) datasets, which contain non-equilibrium structures. However, Wang et al. (2023) does not encode forces, and we show in Section 4.3 that denoising non-equilibrium structures without force encoding can sometimes lead to worse results compared to training without denoising.

### A.2  *SE(3)/E(3)-Equivariant Networks*

We first discuss the concept of equivariance and how equivariant networks achieve equivariance and then compare previous works below. We note that most of the content is adapted from Equiformer (Liao & Smidt, 2023) and EquiformerV2 (Liao et al., 2023). We refer readers to the works (Liao & Smidt, 2023; Liao et al., 2023) for more detailed background on equivariant networks and the work (Duval et al., 2024) for a broader review on geometric GNNs for modeling 3D atomistic systems.

3D atomistic systems are often described in 3D coordinate systems. We have the freedom to choose arbitrary 3D coordinate systems since we can change between different coordinates via the symmetries of 3D space. The relevant 3D symmetris are rotation, translation and inversion. The Euclidean group $E(3)$ consists of 3D rotation, translation and inversion while the special Euclidean group $SE(3)$ is comprised of 3D rotation and translation. The laws of physics remain the same regardless of the coordinate we use, and thus the properties of 3D atomistic systems are equivariant to 3D symmetries. For instance, when a 3D atomistic system is rotated, quantities like energy will remain identical while others like forces will rotate accordingly. Mathematically, a function $f$ mapping between vector spaces $X$ and $Y$ is equivariant to a group of transformations $G$ if for any input $x \in X$, output $y \in Y$ and group element $g \in G$, we have $f(D_X(g)x) = D_Y(g)f(x) = D_Y(g)y$, where $D_X(g)$ and $D_Y(g)$ are transformation matrices or group representations parametrized by $g$ in $X$ and $Y$. Additionally, $f$ is invariant when $D_Y(g)$ is an identity matrix for any $g \in G$.

Incorporating equivariance into neural networks as inductive biases can improve data efficiency and generalizability. Equivariant neural networks (Thomas et al., 2018; Kondor et al., 2018; Weiler et al., 2018; Fuchs et al., 2020; Miller et al., 2020; Townshend et al., 2020; Batzner et al., 2022; Jing et al., 2021; Schütt et al., 2021; Satorras et al., 2021; Brandstetter et al., 2022; Thölke & Fabritiis, 2022; Le et al., 2022; Musaelian et al., 2022; Batatia et al., 2022; Liao & Smidt, 2023; Passaro & Zitnick, 2023; Liao et al., 2023) achieve equivariance to 3D rotation and optionally inversion by using vector spaces of irreducible representations (irreps) as equivariant features. The vector spaces of irreps are $(2L + 1)$-dimensional, with degree $L$ being a non-negative integer. $L$ can be viewed as the angular frequency of vectors and determines how fast vectors change with respect to a rotation of the coordinate system. Vectors of degree $L$ are referred to as type-$L$ vectors. They are transformed with Wigner-D matrices $D^{(L)}$ when rotating the coordinate system, and $D^{(L)}$ of different $L$ acts on independent vector spaces. Euclidean vectors $\vec{r}$ in $\mathbb{R}^3$ such as relative positions and forces can be projected into type-$L$ vectors by using spherical harmonics $Y^{(L)}(\frac{\vec{r}}{||\vec{r}||})$. We concatenate multiple type-$L$ vectors to build an equivariant feature $f$. Given the maximum degree $L_{max}$ of equivariant features, $f$ has $C_L$ type-$L$ vectors, where $0 \leqslant L \leqslant L_{max}$ and $C_L$ is the number of channels for type-$L$ vectors. Both $L_{max}$ and $C_L$ are architectural hyper-parameters of equivariant networks. Equivariant operations are applied to equivariant features to preserve equivariance, and two examples are tensor products and $SO(3)$ linear operations. Tensor products are the fundamental operation in equivariant networks for interacting vectors of different $L$ and are used to build convolutions and attention. On the other hand, $SO(3)$ linear operations apply separate linear operations to each group of type-$L$ vectors. Relevant to the proposed method, we encode forces into equivariant features by first projecting forces to type-$L$ vectors with spherical harmonics and then expanding the number of channels to $C_L$ with an $SO(3)$ linear operation.

Previous works on equivariant networks mainly differ in which equivariant operations are used and the combination of those operations. TFN (Thomas et al., 2018) and NequIP (Batzner et al., 2022) use tensor products for equivariant graph convolution with linear messages, with the latter utilizing extra gate activation (Weiler et al., 2018). SEGNN (Brandstetter et al., 2022) applies gate activation to messages passing

for non-linear messages (Gilmer et al., 2017; Sanchez-Gonzalez et al., 2020). SE(3)-Transformer (Fuchs et al., 2020) adopts equivariant dot product attention with linear messages. Equiformer (Liao & Smidt, 2023) improves upon previous models by combining MLP attention and non-linear messages and additionally introducing equivariant layer normalization and regularizations like dropout (Srivastava et al., 2014) and stochastic depth (Huang et al., 2016). However, these networks rely on compute-intensive $SO(3)$ convolutions built from tensor products, and therefore they can only use small values for maximum degrees $L_{max}$ of irreps features. eSCN (Passaro & Zitnick, 2023) significantly reduces the complexity of $SO(3)$ convolutions by first rotating irreps features based on relative positions and then applying $SO(2)$ linear layers, enabling higher values of $L_{max}$. EquiformerV2 (Liao et al., 2023) adopts eSCN convolutions and proposes an improved version of Equiformer to better leverage the power of higher $L_{max}$, achieving the current state-of-the-art results on OC20 (Chanussot* et al., 2021) and OC22 (Tran* et al., 2022) datasets.

### A.3 Equiformer Series

Equiformer (Liao & Smidt, 2023) is an $SE(3)/E(3)$-equivariant graph neural network that combines the inductive biases of 3D-related equivariance with the strength of Transformers (Vaswani et al., 2017). Starting from Transformers, Equiformer introduces three architectural modifications. First, Equiformer adopts equivariant features built from vector spaces of irreps as internal representations to incorporate equivariance. Second, equivariant operations are applied to the equivariant features. These operations include tensor products and the equivariant counterparts of the original operations in Transformers. The latter part consists of equivariant linear operations (i.e., $SO(3)$ linear operations), equivariant layer normalization (Ba et al., 2016) and gate activation (Weiler et al., 2018). Third, Equiformer proposes to apply non-linear functions to both attention weights and message passing, which improves the expressivity of attention in standard Transformers.

Although Equiformer demonstrates that Transformers generalize well to 3D atomistic systems, it is limited to small values of maximum degree $L_{max}$ because of the compute-intensive tensor product operations. Higher degrees can better capture angular resolutions and directional information, and therefore lower $L_{max}$ can limit the expressivity of Equiformer. To address this limitation, EquiformerV2 (Liao et al., 2023) adopts eSCN (Passaro & Zitnick, 2023) convolutions, which significantly reduce the computational complexity of tensor products, to incorporate higher-degree equivariant representations and proposes three architectural improvements to better leverage the power of higher degrees. First, attention re-normalization is proposed to introduce one additional layer normalization to the non-linear functions of attention weights. This helps stabilize attention and improves empirical performance. Second, separable $S^2$ activation based on $S^2$ activation (Cohen et al., 2018) is proposed to better mix the information of all degrees and stabilize training. Third, separable layer normalization is proposed to replace the original equivariant layer normalization in order to preserve the relative importance of different degrees.

## B Details of Experiments on OC20

### B.1 Training Details

Since each structure in OC20 S2EF dataset has a pre-defined set of fixed and free atoms and we only predict forces of free atoms, we only apply DeNS to free atoms. When partially corrupted structures are used, we add noise to and denoise a random subset of free atoms. When training EquiformerV2 on OC20 S2EF-All+MD dataset, we only apply DeNS to structures from the All split.

For force prediction, we adopt direct methods following previous works for a fair comparison. We add an additional block of equivariant graph attention to EquiformerV2 for noise prediction. We mainly follow the hyper-parameters of training EquiformerV2 without DeNS on OC20 S2EF-2M and S2EF-All+MD datasets. For training EquiformerV2 on OC20 S2EF-All+MD dataset, we increase the number of epochs from 1 to 2 for better performance. This results in higher training time than other methods. However, we note that we already demonstrate training with DeNS can achieve better results given the same amount of training time in Table 1(a). Table 7 summarizes the hyper-parameters of training EquiformerV2 with DeNS for the ablation studies on OC20 S2EF-2M dataset in Section 4.1.1 and for the main results on OC20 S2EF-All+MD

| Hyper-parameters | EquiformerV2 (89M) on OC20 S2EF-2M dataset | EquiformerV2 (160M) on OC20 S2EF-All+MD dataset |
|---|---|---|
| Optimizer | AdamW | AdamW |
| Learning rate scheduling | Cosine learning rate with linear warmup | Cosine learning rate with linear warmup |
| Warmup epochs | 0.1 | 0.01 |
| Maximum learning rate | $2 \times 10^{-4}$ for 12 epochs | $4 \times 10^{-4}$ |
| | $4 \times 10^{-4}$ for 20, 30 epochs | |
| Batch size | 64 for 12 epochs | 512 |
| | 128 for 20, 30 epochs | |
| Number of epochs | 12, 20, 30 | 2 |
| Weight decay | $1 \times 10^{-3}$ | $1 \times 10^{-3}$ |
| Dropout rate | 0.1 | 0.1 |
| Stochastic depth | 0.05 | 0.1 |
| Energy coefficient $\lambda_E$ | 2 | 4 |
| Force coefficient $\lambda_F$ | 100 | 100 |
| Gradient clipping norm threshold | 100 | 100 |
| Model EMA decay | 0.999 | 0.999 |
| Cutoff radius (Å) | 12 | 12 |
| Maximum number of neighbors | 20 | 20 |
| Number of radial bases | 600 | 600 |
| Dimension of hidden scalar features in radial functions $d_{edge}$ | (0, 128) | (0, 128) |
| Maximum degree $L_{max}$ | 6 | 6 |
| Maximum order $M_{max}$ | 2 | 3 |
| Number of Transformer blocks | 12 | 20 |
| Embedding dimension $d_{embed}$ | (6, 128) | (6, 128) |
| $f_{ij}^{(L)}$ dimension $d_{attn\_hidden}$ | (6, 64) | (6, 64) |
| Number of attention heads $h$ | 8 | 8 |
| $f_{ij}^{(0)}$ dimension $d_{attn\_alpha}$ | (0, 64) | (0, 64) |
| Value dimension $d_{attn\_value}$ | (6, 16) | (6, 16) |
| Hidden dimension in feed forward networks $d_{ffn}$ | (6, 128) | (6, 128) |
| Resolution of point samples $R$ | 18 | 18 |
| Probability of optimizing DeNS $p_{\text{DeNS}}$ | 0.5 | 0.125 |
| DeNS coefficient $\lambda_{\text{DeNS}}$ | 10 | 15 |
| Standard deviation of Gaussian noise $\sigma$ | 0.1 for 12 epochs | 0.1 |
| | 0.15 for 20, 30 epochs | |
| Corruption ratio $r_{\text{DeNS}}$ | 0.5 | 0.25 |

Table 7: Hyper-parameters of training EquiformerV2 with DeNS on OC20 S2EF-2M dataset and OC20 S2EF-All+MD dataset.

dataset in Section 4.1.2. Please refer to the work of EquiformerV2 (Liao et al., 2023) for details of the architecture. For training eSCN with DeNS, we use the same DeNS-related hyper-parameters as those for training EquiformerV2 for 20 epochs and the same module as force prediction to predict noise.

V100 GPUs with 32GB are used to train models. We use 16 GPUs for training EquiformerV2 for 12 epochs and eSCN on OC20 S2EF-2M dataset and use 32 GPUs for training EquiformerV2 for 20 and 30 epochs. We train EquiformerV2 with 128 GPUs on OC20 S2EF-All+MD dataset. The training time and the numbers of parameters of different models on OC20 S2EF-2M dataset can be found in Table 1(a). For OC20 S2EF-All+MD dataset, the training time is 88982 GPU-hours and the number of parameters is 160M.

# C  Details of Experiments on OC22

## C.1  Training Details

Different from OC20, all the atoms in a structure in OC22 are free, and we apply DeNS to all the free atoms. We add an additional block of equivariant graph attention to EquiformerV2 for noise prediction. We follow the same practice as on OC20 and use direct methods for force prediction. We follow the same hyper-parameters not relevant to DeNS, and Table 8 summarizes the hyper-parameters for the results on OC22 in Table 3. We use 32 V100 GPUs (32GB) for training. The training time is 5082 GPU-hours, and the number of parameters is 127M.

| Hyper-parameters | Value or description |
|---|---|
| Optimizer | AdamW |
| Learning rate scheduling | Cosine learning rate with linear warmup |
| Warmup epochs | 0.1 |
| Maximum learning rate | $2 \times 10^{-4}$ |
| Batch size | 128 |
| Number of epochs | 6 |
| Weight decay | $1 \times 10^{-3}$ |
| Dropout rate | 0.1 |
| Stochastic depth | 0.1 |
| Energy coefficient $\lambda_E$ | 4 |
| Force coefficient $\lambda_F$ | 100 |
| Gradient clipping norm threshold | 50 |
| Model EMA decay | 0.999 |
| Cutoff radius (Å) | 12 |
| Maximum number of neighbors | 20 |
| Number of radial bases | 600 |
| Dimension of hidden scalar features in radial functions $d_{edge}$ | $(0, 128)$ |
| Maximum degree $L_{max}$ | 6 |
| Maximum order $M_{max}$ | 2 |
| Number of Transformer blocks | 18 |
| Embedding dimension $d_{embed}$ | $(6, 128)$ |
| $f_{ij}^{(L)}$ dimension $d_{attn\_hidden}$ | $(6, 64)$ |
| Number of attention heads $h$ | 8 |
| $f_{ij}^{(0)}$ dimension $d_{attn\_alpha}$ | $(0, 64)$ |
| Value dimension $d_{attn\_value}$ | $(6, 16)$ |
| Hidden dimension in feed forward networks $d_{ffn}$ | $(6, 128)$ |
| Resolution of point samples $R$ | 18 |
| Probability of optimizing DeNS $p_{\text{DeNS}}$ | 0.5 |
| DeNS coefficient $\lambda_{\text{DeNS}}$ | 25 |
| Standard deviation of Gaussian noise $\sigma$ | 0.15 |
| Corruption ratio $r_{\text{DeNS}}$ | 0.5 |

Table 8: Hyper-parameters for OC22 dataset.

# D Details of Experiments on MD17

## D.1 Additional Details of DeNS

It is necessary that gradients consider both the original task and DeNS when updating learnable parameters, and this affects how we sample structures for DeNS when only a single GPU is used for training models on the MD17 dataset. We zero out forces corresponding to structures used for the original task so that a single forward-backward propagation can consider both DeNS and the original task. In contrast, if we switch between DeNS and the original task for different iterations, gradients only consider either DeNS or the original task, and we find that this does not result in better performance on the MD17 dataset than training without DeNS.

## D.2 Training Details

We use the official implementation of Equiformer (Liao & Smidt, 2023) for experiments on the MD17 dataset and follow most of the original hyper-parameters for training with DeNS. Gradient methods are used to predict forces. For training DeNS, we use an additional block of equivariant graph attention for noise prediction, which slightly increases training time and the number of parameters. The hyper-parameters introduced by training DeNS and the values of energy coefficient $\lambda_E$ and force coefficient $\lambda_F$ on different molecules can be found in Table 9. Empirically, we find that linearly decaying DeNS coefficient $\lambda_{\text{DeNS}}$ to 0 thoughout the training can result in better performance. For the Equiformer variant without attention and layer normalization, we find that using normal distributions to initialize weights can result in training divergence and therefore we use uniform distributions. For some molecules, we find training Equiformer variant without attention and layer normalization with DeNS is unstable and therefore reduce the learning rate to $3 \times 10^{-4}$.

We use one A5000 GPU with 24GB to train different models for each molecule. The training time and the numbers of parameters can be found in Table 4.

| Hyper-parameter | Aspirin | Benzene | Ethanol | Malonaldehyde | Naphthalene | Salicylic acid | Toluene | Uracil |
|---|---|---|---|---|---|---|---|---|
| Energy coefficient $\lambda_E$ | 1 | 1 | 1 | 1 | 2 | 1 | 1 | 1 |
| Force coefficient $\lambda_F$ | 80 | 80 | 80 | 100 | 20 | 80 | 80 | 20 |
| Probability of optimizing DeNS $p_{\text{DeNS}}$ | 0.25 | 0.25 | 0.25 | 0.25 | 0.25 | 0.25 | 0.125 | 0.25 |
| DeNS coefficient $\lambda_{\text{DeNS}}$ | 5 | 5 | 5 | 5 | 5 | 5 | 5 | 5 |
| Standard deviation of Gaussian noises $\sigma$ | 0.05 | 0.05 | 0.05 | 0.05 | 0.05 | 0.05 | 0.025 | 0.05 |
| Corruption ratio $r_{\text{DeNS}}$ | 0.25 | 0.25 | 0.25 | 0.25 | 0.25 | 0.25 | 0.25 | 0.25 |

Table 9: Hyper-parameters of training Equiformer ($L_{max} = 2$) and Equiformer ($L_{max} = 3$) with DeNS on the MD17 dataset. Other hyper-parameters not listed here are the same as the original Equiformer trained without DeNS.

| | Aspirin | | | | Ethanol | | | | Naphthalene | | | | Salicylic acid | | | |
|---|---|---|---|---|---|---|---|---|---|---|---|---|---|---|---|---|
| Model | energy↓ | forces↓ | stability↑ | $h(r)$ ↓ | energy↓ | forces↓ | stability↑ | $h(r)$ ↓ | energy↓ | forces↓ | stability↑ | $h(r)$ ↓ | energy↓ | forces↓ | stability↑ | $h(r)$ ↓ |
| Equiformer ($L_{max} = 2$) | 5.3 | 7.2 | **300** | 0.02 | **2.2** | 3.1 | 289.9 | **0.09** | 3.7 | 2.1 | 133.8 | **0.12** | 4.5 | 4.1 | **300** | **0.03** |
| Equiformer ($L_{max} = 2$) + DeNS | **5.1** | **5.7** | **300** | 0.02 | **2.2** | **2.6** | **300** | **0.09** | 3.7 | **1.7** | **157.2** | **0.12** | **4.4** | **3.7** | **300** | **0.03** |

Table 10: Simulation-based results on MD17 dataset. We report simulation-based metrics, stability and distribution of interatomic distances $h(r)$. Models are the same as in Table 4. Energy and force are in units of meV and meV/Å and are evaluated on the testing set.

### D.3 Additional Simulation-Based Results

Following the work (Fu et al., 2023), we run simulations on the four molecules (i.e., Aspirin, Ethanol, Naphthalene and Salicylic Acid) and compare the two simulation-based metrics, which are stability and distribution of interatomic distances $h(r)$. We use the previously trained Equiformer for the simulations. We note that Equiformer models are trained on 950 examples for each molecule instead of $9,500$ as in Fu et al. (2023). The results are summarized in Table 10. Training with DeNS helps energy and forces MAE as well as stability. The results of $h(r)$ are similar. For Naphthalene, training with DeNS improves stability from $133.8/300$ to $157.2/300$. Both models become unstable when running simulations of Naphthalene, and we surmise that is because we only use 950 examples for training instead of $9,500$. For others, the performance gain in simulation-based metrics is not significant since the original Equiformer trained on 950 examples already achieves similar results to other top-performing models trained on $9,500$ examples, suggesting that the potential room for improvement would be quite limited.

## E Pseudocode for Force Encoding

We provide the pseudocode for force encoding in Algorithm 1. Here the original node embedding $x_i$ contains only the atom embedding $x_{i,z}$. Note that the type-$L$ vectors in $x_{i,z}$ are all zeros for $L > 0$ since $x_{i,z}$ is obtained by applying an $SO(3)$ linear layer to type-0 vectors. We directly add the force embedding $x_{i,\mathbf{f}}$ to the original node embedding to encode forces.

---
**Algorithm 1** Force Encoding
---
1: $x_{i,z} \leftarrow$ SO3_Linear $\left(\text{one\_hot}(z_i)\right)$      ▷ Extend Equation 5 to all degrees $L$ and apply to the one-hot encoding of atomic numbers $z_i$ for each atom
2: $x_i \leftarrow x_{i,z}$
3: $x_{i,\mathbf{f}} \leftarrow$ SO3_Linear $\left(||\mathbf{f}_i|| \cdot Y\left(\frac{\mathbf{f}_i}{||\mathbf{f}_i||}\right)\right)$
4: $x_i \leftarrow x_i + x_{i,\mathbf{f}}$
---

## F Pseudocode for Training with DeNS

We provide the pseudocode for training with DeNS in Algorithm 2 and note that Line 5 can be parallelized. For denoising partially corrupted structures discussed in Section 3.2.3, we only add noise to a random subset of atoms (Line $11 - 14$) and predict the corresponding noise (Line 25).

---

**Algorithm 2** Training with DeNS

---

1: **Input:**
$p_{\text{DeNS}}$: probability of optimizing DeNS
$\lambda_{\text{DeNS}}$: DeNS coefficient
$\sigma$: standard deviation of Gaussian noise
$r_{\text{DeNS}}$: corruption ratio
$\lambda_E$: energy coefficient
$\lambda_F$: force coefficient
GNN: graph neural network for predicting energy, forces and noise
2: **while** training **do**
3: $\quad \mathcal{L}_{\text{total}} = 0$
4: $\quad$ Sample a batch of $B$ structures $\{(S_{\text{non-eq}})^j \mid j \in \{1, ..., B\}\}$ from the training set
5: $\quad$ **for** $j = 1$ to $B$ **do** $\qquad\qquad\qquad\qquad\qquad\qquad\qquad$ ▷ This for loop can be parallelized
6: $\qquad$ Let $(S_{\text{non-eq}})^j = \left\{ (z_i, \mathbf{p}_i) \mid i \in \{1, ..., |(S_{\text{non-eq}})^j|\} \right\}$
7: $\qquad$ Sample $p$ from a uniform distribution $\mathbf{U}(0,1)$ to determine whether to optimize DeNS
8: $\qquad$ **if** $p < p_{\text{DeNS}}$ **then** $\qquad\qquad\qquad\qquad\qquad$ ▷ Optimize DeNS based on Equation 6
9: $\qquad\quad$ **for** $i = 1$ to $|(S_{\text{non-eq}})^j|$ **do**
10: $\qquad\qquad$ $q_i \sim \mathbf{U}(0,1)$
11: $\qquad\qquad$ **if** $q_i < r_{\text{DeNS}}$ **then** $\qquad\qquad\qquad\qquad$ ▷ Add noise to and denoise the atom
12: $\qquad\qquad\quad$ $\boldsymbol{\epsilon}_i \sim \mathcal{N}(0, \sigma I_3)$
13: $\qquad\qquad\quad$ $\tilde{\mathbf{p}}_i = \mathbf{p}_i + \boldsymbol{\epsilon}_i$
14: $\qquad\qquad\quad$ $m_i = 1$ $\qquad\qquad\qquad\qquad$ ▷ Denoise the atom when calculating $\mathcal{L}_{\text{DeNS}}$
15: $\qquad\qquad$ **else**
16: $\qquad\qquad\quad$ $\tilde{\mathbf{p}}_i = \mathbf{p}_i$
17: $\qquad\qquad\quad$ $m_i = 0$
18: $\qquad\qquad$ **end if**
19: $\qquad\qquad$ $\tilde{\mathbf{f}}'_i = \mathbf{f}'_i \cdot m_i$ $\qquad\qquad\qquad$ ▷ Encode the atomic force if the atom is corrupted
20: $\qquad\quad$ **end for**
21: $\qquad$ Let $(\tilde{S}_{\text{non-eq}})^j = \left\{ (z_i, \tilde{\mathbf{p}}_i) \mid i \in \{1, ..., |(S_{\text{non-eq}})^j|\} \right\}$
22: $\qquad$ Let $\tilde{F}\left((S_{\text{non-eq}})^j\right) = \left\{ \tilde{\mathbf{f}}'_i \mid i \in \{1, ..., |(S_{\text{non-eq}})^j|\} \right\}$
23: $\qquad$ $\hat{E}, \hat{F}, \hat{\boldsymbol{\epsilon}} \leftarrow \text{GNN}\left((\tilde{S}_{\text{non-eq}})^j, \tilde{F}\left((S_{\text{non-eq}})^j\right)\right)$
24: $\qquad$ $\mathcal{L}_E = \left| E'\left((S_{\text{non-eq}})^j\right) - \hat{E} \right|$ $\qquad\qquad\qquad$ ▷ Predict energy of the original structure
25: $\qquad$ $\mathcal{L}_{\text{DeNS}} = \frac{1}{|(S_{\text{non-eq}})^j|} \sum_{i=1}^{|(S_{\text{non-eq}})^j|} m_i \cdot \left| \frac{\boldsymbol{\epsilon}_i}{\sigma} - \hat{\boldsymbol{\epsilon}}_i \right|^2$ $\qquad$ ▷ Predict noise of corrupted atoms
26: $\qquad$ $\mathcal{L}_F = \frac{1}{|(S_{\text{non-eq}})^j|} \sum_{i=1}^{|(S_{\text{non-eq}})^j|} (1 - m_i) \cdot |\mathbf{f}'_i\left((S_{\text{non-eq}})^j\right) - \hat{\mathbf{f}}_i|^2$ $\quad$ ▷ Predict forces of uncorrupted atoms
27: $\qquad$ $\mathcal{L}_{\text{total}} = \mathcal{L}_{\text{total}} + \lambda_E \cdot \mathcal{L}_E + \lambda_{\text{DeNS}} \cdot \mathcal{L}_{\text{DeNS}} + \lambda_F \cdot \mathcal{L}_F$
28: $\qquad$ **else** $\qquad\qquad\qquad\qquad\qquad\qquad\qquad$ ▷ Optimize the original task based on Equation 1
29: $\qquad\quad$ $\hat{E}, \hat{F}, \_\_ \leftarrow \text{GNN}\left((S_{\text{non-eq}})^j\right)$
30: $\qquad\quad$ $\mathcal{L}_E = \left| E'\left((S_{\text{non-eq}})^j\right) - \hat{E} \right|$
31: $\qquad\quad$ $\mathcal{L}_F = \frac{1}{|(S_{\text{non-eq}})^j|} \sum_{i=1}^{|(S_{\text{non-eq}})^j|} |\mathbf{f}'_i\left((S_{\text{non-eq}})^j\right) - \hat{\mathbf{f}}_i|^2$
32: $\qquad\quad$ $\mathcal{L}_{\text{total}} = \mathcal{L}_{\text{total}} + \lambda_E \cdot \mathcal{L}_E + \lambda_F \cdot \mathcal{L}_F$
33: $\qquad$ **end if**
34: $\quad$ **end for**
35: $\quad$ $\mathcal{L}_{\text{total}} = \frac{\mathcal{L}_{\text{total}}}{B}$
36: $\quad$ Optimize GNN based on $\mathcal{L}_{\text{total}}$
37: **end while**

## G  Visualization of Corrupted Structures

We visualize how adding noise of different scales affects structures in OC20, OC22 and MD17 datasets in Figure 3, Figure 4 and Figure 5, respectively.

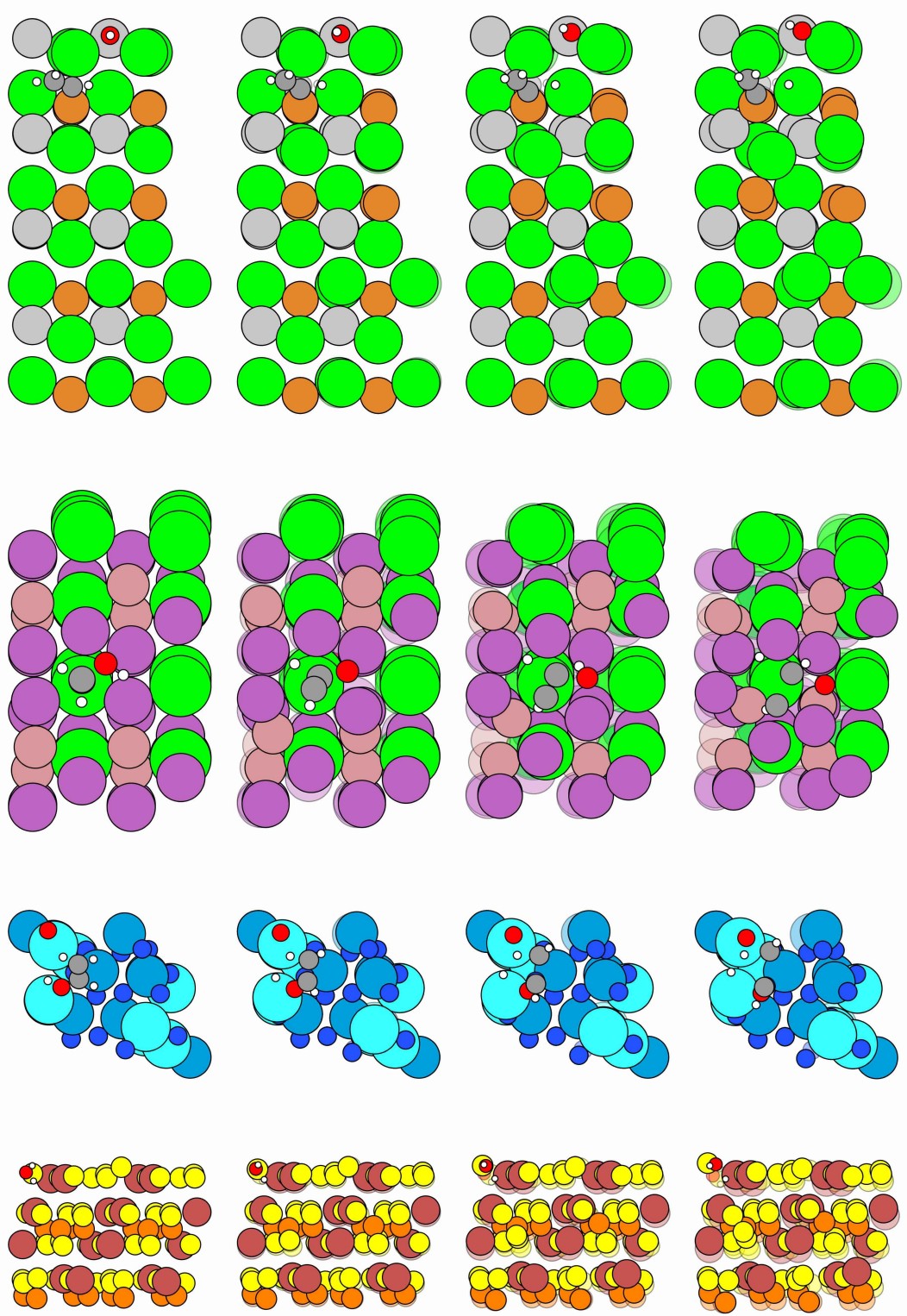

Figure 3: Visualization of corrupted structures in OC20 dataset. We add noise of different scales to original structures (column 1). For each row, we sample $\epsilon_i \sim \mathcal{N}(0, I_3)$, multiply $\epsilon_i$ with $\sigma = 0.1$ (column 2), $0.3$ (column 3) and $0.5$ (column 4), and add the scaled noise to the original structures. For columns 2, 3 and 4, the lighter colors denote the atomic positions of the original structures. Here we add noise to all the atoms in a structure for better visual effects.

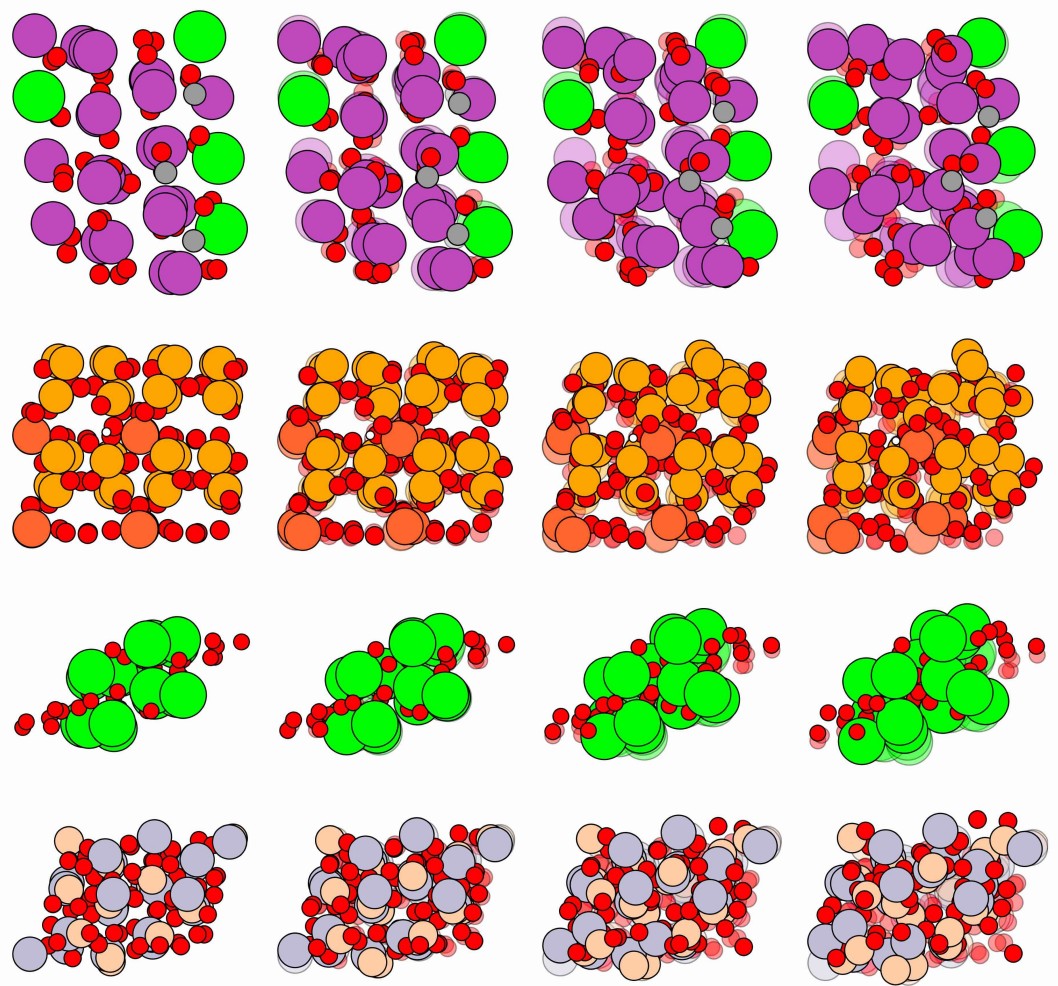

Figure 4: Visualization of corrupted structures in OC22 dataset. We add noise of different scales to original structures (column 1). For each row, we sample $\epsilon_i \sim \mathcal{N}(0, I_3)$, multiply $\epsilon_i$ with $\sigma = 0.1$ (column 2), $0.3$ (column 3) and $0.5$ (column 4), and add the scaled noise to the original structures. For columns 2, 3 and 4, the lighter colors denote the atomic positions of the original structures. Here we add noise to all the atoms in a structure for better visual effects.

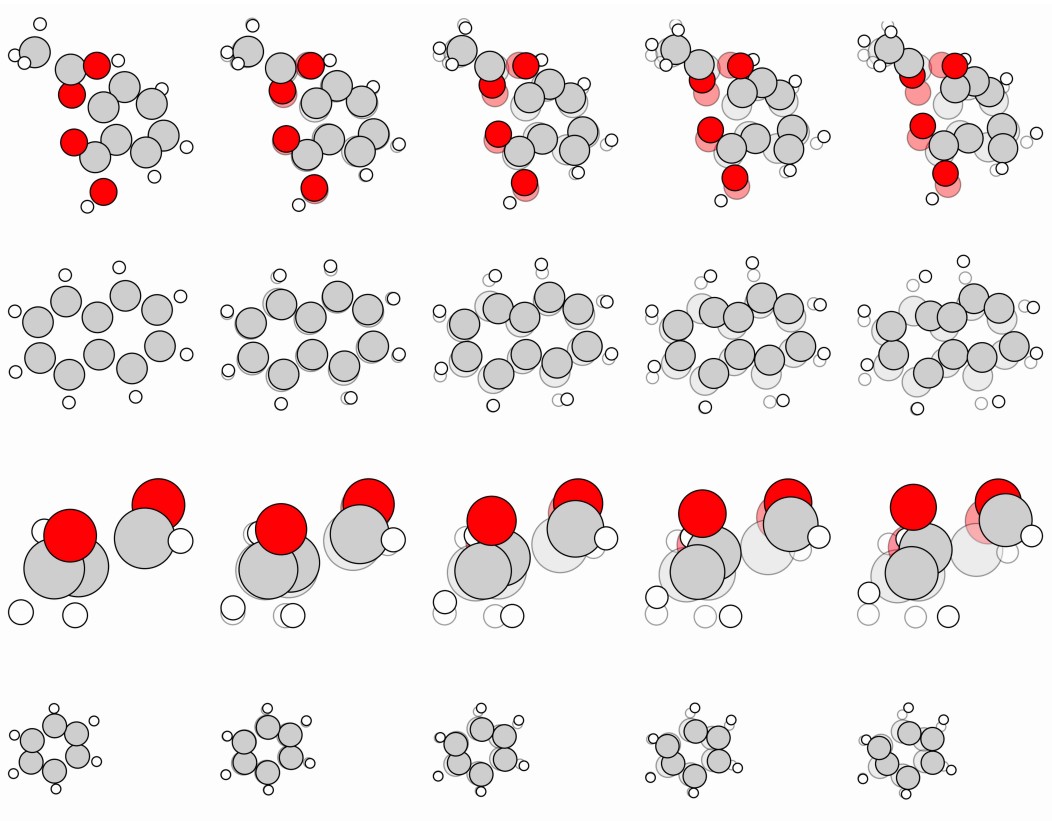

Figure 5: Visualization of corrupted structures in MD17 dataset. We add noise of different scales to original structures (column 1). For each row, we sample $\epsilon_i \sim \mathcal{N}(0, I_3)$, multiply $\epsilon_i$ with $\sigma = 0.01$ (column 2), $0.03$ (column 3), $0.05$ (column 4) and $0.07$ (column 5), and add the scaled noise to the original structures. For columns 2, 3, 4 and 5, the lighter colors denote the atomic positions of the original structures. Here we add noise to all the atoms in a structure for better visual effects.

