# OpenReview forum: "Generalizing Denoising to Non-Equilibrium Structures Improves Equivariant Force Fields"
_TMLR — Accepted by TMLR_

### Review · Reviewer_rRiX · 2024-09-11

**Summary Of Contributions:**

The paper proposes a new technique for training of equivariant networks called denoising non-equilibrium structures (DeNS). The key differentiator in DeNS is that it enables training of equivariant networks on non-equilibrium atomistic structures by using the force as an additional feature for denoising. Including the force makes the problem no longer ill-posed as the structure to be denoised is better identified.

The paper starts by providing an introduction and motivation in Section 1, followed by a discussion of related works in Section 2. Both sections focus on prior denoising works and geometric deep learning architectures. Section describes the main method, including the setup of DeNS, the DeNS force encoding and the formulation of DeNS as auxiliary loss for training models on energy and force prediction tasks. Section 3 also includes a discussion on the reasons behind why DeNS appears to be beneficial in training equivariant networks focused on data augmentation, connections to self-supervised learning and marginal training time increases.

Section 4 outlines the main experiments of the paper which focus primarily on training EquiformerV2 with DeNS and understanding the difference in performance on OC20, OC22 and MD17. The results generally show that DeNS is helpful for OC20 and OC22 with some more mixed results on MD17. Section 5 ends the paper with a conclusion.

**Audience:**

Yes

**Claims And Evidence:**

Yes

**Requested Changes:**

**Requested Changes**
* Provide more detail on how the force of the disturbed structure is calculated and/or provided as input.
* Discussion of additional related work, including a review of atomistic modeling using GNNs [1]. It would also be interesting to discuss arriving at equivariance via data augmentations [2] as those methods appear to be getting more attention due to their lower computational costs. The paper could also discuss additional datasets [3] and modeling settings [4] where DeNS could be relevant.


**Nice to Have**
* Did you do any ablation on equilibrium-based training compared to DeNS?
* Do you think DeNS has any effects on the learning manifold? Prior work has tried to visualize energy landscapes to understand this further [5]. Maybe this can provide further details on why partially perturbed structures work better.
* You briefly mentioned that equivariant models are most suited because forces are an equivariant quantity. What do you think would be needed to extend to other modeling methods? Do you think DeNS could be useful in setting with different materials representations? [6] [7]



[1] Duval, Alexandre, et al. "A Hitchhiker's Guide to Geometric GNNs for 3D Atomic Systems." arXiv preprint arXiv:2312.07511 (2023).

[2] Duval, Alexandre Agm, et al. "Faenet: Frame averaging equivariant gnn for materials modeling." International Conference on Machine Learning. PMLR, 2023.

[3] Jain, Anubhav, et al. "Commentary: The Materials Project: A materials genome approach to accelerating materials innovation." APL materials 1.1 (2013).

[4] Lee, Kin Long Kelvin, et al. "Towards foundation models for materials science: The open matsci ml toolkit." Proceedings of the SC'23 Workshops of The International Conference on High Performance Computing, Network, Storage, and Analysis. 2023.

[5] Bihani, Vaibhav, et al. "Low-Dimensional Projections for Visualizing Energy Landscapes of Atomic Systems." AI for Accelerated Materials Design-Vienna 2024.

[6] Damewood, James, et al. "Representations of materials for machine learning." Annual Review of Materials Research 53.1 (2023): 399-426.

[7] Alampara, Nawaf, Santiago Miret, and Kevin Maik Jablonka. "MatText: Do Language Models Need More than Text & Scale for Materials Modeling?." AI for Accelerated Materials Design-Vienna 2024.

**Strengths And Weaknesses:**

**Strengths:**
* The paper presents a practical auxiliary training method for equivariant models on atomistic modeling. DeNS has the potential to be applied to diverse modeling problems as an auxiliary loss term.
* The paper describes the method formulation in great detail and provides a detailed discussion of prior work on equivariant networks and denoising approaches.
* The experimental analysis on Equiformer-V2 is quite detailed for the datasets studied.

**Weaknesses:**
* The paper could be further strengthened with some additional discussion on how DeNS could be applied to other types of modeling settings. Since the authors mention that partially corrupted structures often train better, it might be good to have a discussion on how DeNS might be practically applied to other situations.

---

> ### Author Response · Authors · 2024-10-18
> **Response to Reviewer rRiX (1/3)**
>
> We thank the reviewer for helpful feedback and address the comments below. In addition, we have updated our manuscript based on the comments and highlighted the differences in blue.
>
> ---
>
> > 1. [Weakness 1] some additional discussion on how DeNS could be applied to other types of modeling settings.
>
> Please see 5. and 9. for the additional discussions on different modeling settings.
>
> ---
>
> > 2. [Weakness 2] it might be good to have a discussion on how DeNS might be practically applied to other situations.
>
> DeNS can also be applied to other quantities, which might specify a unique but not necessarily equilibrium structure, such as forces, local polarization, charge density and so on. Besides, we discuss how DeNS can be applied to different datasets in 6. Please let us know if this addresses the comment of “other situations”.
>
> ---
>
> > 3. [Requested Changes 1] Provide more detail on how the force of the disturbed structure is calculated and/or provided as input.
>
> We are not sure what the specific details are and interpret the question in many ways as below.
>
> We re-iterate the details of force encoding when denoising corrupted structures mentioned in the paper. First, we reuse the force labels in the training set without any additional labeling (Figure 2) but treat them as inputs (Figure 2(b) and 2(c)). The target of denoising is the displacement vector needed to achieve the desired input force on a corrupted atom. Second, when denoising partially corrupting structures (Figure 2(c)), we encode forces and predict noise for corrupted atoms while we predict forces without encoding forces for uncorrupted atoms. Third, when encoding forces for corrupted atoms, we need to consider information of both magnitude and direction. For equivariant networks based on irreducible representations (irreps), the internal representations (e.g., node embeddings) consist of vectors of degrees from 0 to $L_{max}$. Since forces are vectors of degree 1, we can simply expand forces to vectors of degrees up to $L_{max}$ to obtain force embedding (Section 3.2.2) and then add the force embedding to the initial node embedding.
>
> Additionally, we provide the pseudocode for encoding forces along with atom types in node embeddings as below and will add this pseudocode to Section E in appendix.
>
> $z_i, f_i$ $←$ get_atom_type($S$), get_force($S$)
>
> atom_embedding $←$ SO(3)_Linear_atom_type($z_i$)
>
> force_embedding $←$ SO(3)_Linear_force($Y(f_i / || f_i ||)$) $* || f_i ||$  # Equation (5)
>
> node_embedding $←$ atom_embedding + force_embedding
>
> ---
>
> > 4. [Requested Changes 2] Discussion of additional related work, including a review of atomistic modeling using GNNs [1].
>
> We will include a reference to this review on geometric GNNs at the end of Section 2.2. However, due to the huge amount of works mentioned in the review, we would not be able to discuss them here. Please let us know if there is any specific work that you find relevant and should be discussed here.
>
> ---
>
> > 5. [Requested Changes 2] Discussion on arriving at equivariance via data augmentations [2].
>
> We give a brief overview of their method [2] and discuss how to encode forces here.
>
> [2] proposed to use frame averaging to convert 3D atomistic systems to canonical representations and apply unconstrained functions to the canonical representations. Specifically, they first perform Principal Component Analysis (PCA) to 3D atomic positions and select the three principal components with the highest eigenvalues as one set of frame axes. Then, they project the 3D atomic positions to the frame axes to obtain the canonical representation, which remains identical under arbitrary $E(3)$ transformations and therefore allows using any learnable function. Since there are 8 different choices of frame axes, the above projection will be repeated for 8 times for exact equivariance or invariance. They show that randomly sampling 1 set of frame axes, which can be viewed as a variant of data augmentation, and letting models learn the symmetries of other sets during training are sufficient to achieve decent results.
>
> To encode forces when optimizing for DeNS, we can follow how unit cell Cartesian coordinates are projected in their framework. Specifically, we compute the set of frame axes using only 3D atomic positions and project the input forces to the frame axes. Same as the projected 3D atomic positions, the projected forces remain the same under any $E(3)$ transformation and enable using unconstrained functions to encode the input forces. We will add the discussion on how [2] can encode forces to the end of Section 3.2.2.
>
> We appreciate the comment on discussing other models and leave testing those models as future work.

---

> > ### Author Response · Authors · 2024-10-18
> > **Response to Reviewer rRiX (2/3)**
> >
> > > 6. [Requested Changes 2] The paper could also discuss additional datasets [3] and modeling settings [4] where DeNS could be relevant.
> >
> > The proposed DeNS can be directly applied to Materials Project [3] as well. Similar to OC20 and OC22 datasets, for each entry in the Materials Project dataset, the Materials Project Trajectory (MPtrj) dataset contains the relaxation trajectory between initial and relaxed structures. Most structures in MPtrj are non-equilibrium and have energy and force labels. Therefore, we can apply DeNS to MPtrj dataset in the same way as OC20 and OC22 datasets. We note that during the period of rebuttal, this work (https://arxiv.org/abs/2410.12771) applies the proposed DeNS to EquiformerV2 and achieves the current best results on Matbench Discovery (https://matbench-discovery.materialsproject.org/), showing the generality of DeNS.
> >
> > For the modeling settings in [4], they used EGNN, which belongs to equivariant networks. Therefore, the proposed method can be directly applied to [4]. Please let us know if our understanding is not correct.
> >
> > ---
> >
> > > 7. [Nice to Have 1] Ablation on equilibrium-based training compared to DeNS?
> >
> > We are not sure what equilibrium-based training refers to here. DeNS without force encoding is the same as the methodology applied to denoising equilibrium structures, and we have shown that it does not work for non-equilibrium structures (Index 3 in Table 1(b)). Besides, if the original structure is at equilibrium with near-zero forces, the proposed method reduces to Noisy Nodes (https://arxiv.org/abs/2106.07971), which has demonstrated its effectiveness on datasets containing only equilibrium structures. Please let us know if this addresses your question.
> >
> > ---
> >
> > > 8. [Nice to Have 2] Effects on the learning manifold? Prior work has tried to visualize energy landscapes to understand this further [5]. Maybe this can provide further details on why partially perturbed structures work better.
> >
> > Yes, the proposed method should affect the learning manifold in the below manner. Given two consecutive structures S1 and S2 in a relaxation trajectory, since we add noise to both S1 and S2 during training, models would see some structures interpolated between S1 and S2. Those interpolated structures would smooth the predictions of models given structures around S1 or S2 as inputs. In contrast, if the models are not trained on those interpolated structures, their outputs can have more variation and be less smooth when taking structures around S1 or S2 as inputs.
> >
> > [5] proposes to visualize energy landscapes in the following manner. First, given a reference structure, they use either optimization trajectories or molecular dynamics simulations (Section 3 in [5]) to perturb the reference structure and obtain some neighboring structures. The neighboring structures form an energy landscape. Second, they compute the relative position vectors between the reference structure and all the neighboring structures and stack them into a matrix said A. Third, they perform principal component analysis to the matrix A and choose the two principal components with the largest eigenvalues as the projected directions said alpha and beta. By projecting the high-dimensional atomic positions (3 times numbers of atoms) to the low-dimensional alpha and beta, they can easily visualize how energy changes with alpha and beta. However, based on their work, it is unclear how projecting to lower dimensions affects the original high-dimensional energy landscapes.
> >
> > Instead, following what we discussed in the paper, we believe that partially corrupted structures can (1) mitigate the issue of making denoising too difficult and potentially not well-defined and (2) generate structures closer to the original data distribution. Both explain why intuitively DeNS with partially corrupted structures can be better than DeNS without partially corrupted structures.

---

> > > ### Author Response · Authors · 2024-10-18
> > > **Response to Reviewer rRiX (3/3)**
> > >
> > > > 9. [Nice to Have 3] What do you think would be needed to extend to other modeling methods? Do you think DeNS could be useful in settings with different materials representations? [6] [7]
> > >
> > > We think DeNS can be directly applied to different models with different representations (e.g., equivariant features, invariant features or texts) as long as they can encode the information of 3D atomic positions and input forces. The only extension required is how to encode forces into those modeling methods to make denoising non-equilibrium structures tractable.
> > >
> > > [6] mainly discusses representations like local descriptors, global descriptors and compositions in addition to equivariant features. For local and global descriptors, they extract distances and angles from 3D atomic positions. Therefore, we can directly follow what we have discussed in the paper to encode forces (the last paragraph in Section 3.2.2) and apply DeNS without any other modification. For representations using only compositions without any 3D atomic positions, we cannot apply DeNS since we add noise to 3D atomic positions, and those representations cannot be used to predict forces either. We note that the representations discussed above achieve much worse performance compared to using equivariant features given 3D atomic positions [6].
> > >
> > > [7] discusses different ways to represent materials as texts and evaluates how well language models can model materials given those texts. For some representations not considering 3D atomic positions, DeNS cannot be applied since we are adding noise to 3D atomic positions, and the models cannot deal with force prediction. For others, we can append input forces after atomic positions and atom types as texts in a similar manner as how they encode atomic positions so that the language models can consider the information of forces as inputs when denoising.

---

> > > > ### Comment · Reviewer_rRiX · 2024-11-13
> > > >
> > > > Thank you for addressing the questions and feedback. While a discussion on the additional types of representations would have been nice (as it can help clarify potential limitations), the discussion in the paper and appendix is already quite extensive. With the changes made based on reviewer feedback, I am supportive of the paper.

---

### Review · Reviewer_7SRu · 2024-09-20

**Summary Of Contributions:**

The paper proposes DeNS (Denoising Non-equilibrium Structures), an auxiliary task for neural networks in 3D atomistic systems, aimed at better leveraging limited training data. It also offers crucial insights into the challenges of denoising non-equilibrium structures, particularly addressing the ambiguity stemming from multiple possible target structures for a given input and propose a novel solution by incorporating force embeddings

The efficacy of method is evaluated through comprehensive evaluations, with significant improvements in both force and energy predictions across OC20, OC22, and MD17 datasets.

**Audience:**

Yes

**Claims And Evidence:**

Yes

**Requested Changes:**

Clarifying the points in the previous section.

**Strengths And Weaknesses:**

### Strengths
- The paper is quite clear in its setup and execution.
- Insights into the challenges of denoising non-equilibrium structures.
- Improvements in force and energy predictions across different datasets.

### Weaknesses and Questions
- Given the existence of multiple equilibrium structures (e.g., molecular conformers), can the model handle situations where a corrupted non-equilibrium structure is more closer to a different equilibrium structure, leading to non-consistent denoising?
- Are the ground truth forces also used during the evaluation time as an encoding for corrupted atoms to predict the ground truth forces and energy?
- How can the above method be adapted for real-world scenarios where force and energy estimation is required for a given non-equilibrium structure without prior knowledge of its forces, a context in which models like MACE and NequIP typically operate?
- Do you create one corrupted structure per molecule or many while training?

---

> ### Author Response · Authors · 2024-10-18
> **Response to Reviewer 7SRu**
>
> We thank the reviewer for helpful feedback and address the comments below.
>
> ---
>
> > 1. [Weaknesses and Questions 1] Given the existence of multiple equilibrium structures (e.g., molecular conformers), can the model handle situations where a corrupted non-equilibrium structure is more closer to a different equilibrium structure, leading to non-consistent denoising?
>
> Yes, using force encoding can handle such situations. Given a corrupted structure, we will always take the corresponding forces of its original structure as input and predict the structure satisfying the input forces. Therefore, even if a given corrupted non-equilibrium structure is closer to a different equilibrium one, we will predict a non-equilibrium structure satisfying input forces rather than a nearby equilibrium structure with forces close to zero.
>
> ---
>
> > 2. [Weaknesses and Questions 2] Are the ground truth forces also used during the evaluation time as an encoding for corrupted atoms to predict the ground truth forces and energy?
>
> No, we apply DeNS to structures (and therefore use ground truth forces) only from the training set. For evaluation on the validation and test sets, we do not apply DeNS and do not need any force labels. We discussed this in detail in Figure 2 and Section 3.2.3.
>
> ---
>
> > 3. [Weaknesses and Questions 3] How can the above method be adapted for real-world scenarios where force and energy estimation is required for a given non-equilibrium structure without prior knowledge of its forces, a context in which models like MACE and NequIP typically operate?
>
> Yes, the proposed method aims at force and energy prediction. Same as the response in 2. and how other models like MACE and NequIP are trained, we only use force labels from the training set and do not require any prior knowledge of forces when evaluating on validation and test sets. In fact, the proposed method should be viewed as augmenting training objectives with denoising to better leverage training data and does not affect evaluation.
>
> ---
>
> > 4. [Weaknesses and Questions 4] Do you create one corrupted structure per molecule or many while training?
>
> We create one random corrupted structure when sampling each structure from a training set (Algorithm 1 in the previous version and Algorithm 2 in the newer version). If we train for many epochs and sample the same structure several times, different random noise will be added each time the structure is sampled.

---

> > ### Author Response · Authors · 2024-11-20
> > **Please let us know if you have other questions or comments**
> >
> > We believe we have addressed your concerns.
> > Please let us know if you have other questions or comments. We will be very happy to respond.

---

### Review · Reviewer_q8wm · 2024-10-29

**Summary Of Contributions:**

This manuscript introduces Denoising Non-Equilibrium Structures (DeNS) as a novel spin on the popular auxiliary task of training models to denoising equilibrium structures to improve atomistic models' performance on tasks like energy and force prediction. The primary contribution is the extension of denoising approaches from equilibrium to non-equilibrium structures, which comprise the majority of atomic data but lack a unique target structure due to the presence of non-zero forces. This approach utilizes the forces from the original, unperturbed structure to guide denoising and reduce the inherent ambiguity in denoising non-equilibrium structures. Due to the requirement of explicitly encoding and representing the forces of the original structure, this approach is particularly well-suited to neural networks that are equivariant to the 3D roto-translational group SO(3).
The authors present experiments on a variety of datasets such as OC20, OC22, and MD17, showing that this approach simultaneously improves performance while reducing training time for energy and force prediction for non-equilibrium structures.

**Audience:**

Yes

**Broader Impact Concerns:**

No broader impact concerns to note.

**Claims And Evidence:**

Yes

**Requested Changes:**

- A brief explanation of the concept of equivariance and equivariant models' foundational principles and basic operations would aid accessibility for readers less familiar with the topic. If space is constrained, these details could be included in an appendix.
- Additional details on the Equiformer model's design would make the paper more self-contained, this information should ideally be included in the appendix.
- Some additional evaluations based on ML generated molecular dynamics trajectories, such as stability and distribution of interatomic distances, would definitely be useful and add a further use-case relevant demonstration of the untility of the DeNS approach.
- Most forces prediction models tend to obtain the forces as the analytical derivative of the energies, which guarantees energy conservation. While the alternative approach taken by the authors has become more common as of late, the authors still need to discuss and justify their choice of directly predicting the forces, thus not guaranteeing energy conservation. Commenting on whether this non-conservative approach might extend to models with explicitly energy-conserving forces could also be useful for other readers.

**Strengths And Weaknesses:**

Strengths:
- The authors have the great insight of recognizing that the force information is already present in denoising tasks for equilibrium structures, and, leverage the forces of non-equilibrium conformers effectively expand the denoising task to non-equilibrium structures, helping to disentangle the ambiguity associated with non-equilibrium target structures.
- It is well-written and concise, presenting a clear and focused narrative on the DeNS approach definition, function and resulting improvements.
- The experiments are thorough, including detailed ablation studies that separate the effects of each design choice, adding to the robustness of the findings.
- Applying DeNS to the Equiformer model results in improvements across multiple datasets, enhancing both prediction accuracy and training efficiency, which is a great achievement.

Weaknesses:
- The main weakness of this work is the lack of any benchmarks beyond energy or force errors. It is well known that demonstrating low. errors on energies and forces for test structures does not tell the full picture for models that are to be used for MD simulations or optimization. It would be very important to see whether DeNS improves the performance in any of the MD-related metrics such as stability and distribution of interatomic distances.
- Other more minor complaints are the very short introduction about equivariant models in general, and the lack of information about the Equiformer model architecture.

---

> ### Author Response · Authors · 2024-11-06
> **Response to Reviewer q8wm**
>
> We thank the reviewer for helpful feedback and address the comments below.
>
> ---
>
> > 1. [Weaknesses 1] The main weakness of this work is the lack of any benchmarks beyond energy or force errors. It is well known that demonstrating low errors on energies and forces for test structures does not tell the full picture for models that are to be used for MD simulations or optimization.
>
> We mentioned this work mainly focuses on energy and force predictions in the main text and the title. In addition, we also considered IS2RE (initial structure to relaxed energy), which includes first relaxing a non-equilibrium structure with ML forces and then predicting the energy of the relaxed structure with the ML model. The results were in Table 2 (OC20) and Table 3 (OC22), and the proposed method further improves the previous state-of-the-art results. Therefore, the proposed DeNS can be helpful to relaxing structures or geometry optimization.
>
> Moreover, during the period of rebuttal, this work (https://arxiv.org/abs/2410.12771) applies the proposed DeNS to EquiformerV2 and demonstrates that DeNS can also improve other metrics (e.g., stress in the first two rows in Table 9 and F1 score and other metrics in Table 5).
>
> ---
>
> > 2. [Weakness 1]  It would be very important to see whether DeNS improves the performance in any of the MD-related metrics such as stability and distribution of interatomic distances.
>
> We highly appreciate this valuable comment from the reviewer but think evaluating MD-related metrics can be left as future work but not necessary in this work. We note that energy, forces and relaxed energy are much better benchmarked compared to MD-related metrics and therefore improving those metrics on all the three datasets in this paper can show the effectiveness of the proposed method. Additionally, the previous works mentioned in the paper did not benchmark on MD-related metrics, which can make a fair comparison difficult.
>
> ---
>
> > 3. [Weakness 2] Short introduction about equivariant models in general, and the lack of information about the Equiformer model architecture.
>
> We do not put much emphasis on the introduction to equivariant networks since this work is about better training methods, not model design. The proposed method can also be applied to models other than equivariant networks. Instead, we did provide the pointers to specific related works for detailed background on equivariant networks (Section A.2) and Equiformer (Section B.1).
>
> ---
>
> > 4. [Requested Changes 1] A brief explanation of the concept of equivariance and equivariant models' foundational principles and basic operations would aid accessibility for readers less familiar with the topic. If space is constrained, these details could be included in an appendix.
>
> Please see 3. We provide more details in Section A.2. Please let us know if there is any additional point that would be great to include.
>
> ---
>
> > 5. [Requested Changes 2] Additional details on the Equiformer model's design would make the paper more self-contained, this information should ideally be included in the appendix.
>
> We provide a summary of the Equiformer series in Section A.3. Please let us know if any further detail is required.
>
> ---
>
> > 6. [Requested Changes 3] Some additional evaluations based on ML generated molecular dynamics trajectories, such as stability and distribution of interatomic distances, would definitely be useful and add a further use-case relevant demonstration of the utility of the DeNS approach.
>
> Please see 2.
>
> ---
>
> > 7. [Requested Changes 4] Most force prediction models tend to obtain the forces as the analytical derivative of the energies, which guarantees energy conservation. While the alternative approach taken by the authors has become more common as of late, the authors still need to discuss and justify their choice of directly predicting the forces, thus not guaranteeing energy conservation. Commenting on whether this non-conservative approach might extend to models with explicitly energy-conserving forces could also be useful for other readers.
>
> As mentioned in “Main Results” in Section 4.3, we did consider both direct methods (OC20 and OC22) and gradient methods (MD17) for force prediction. We simply follow the practice of previous works for a fair comparison and see whether the proposed DeNS can improve the existing models. For OC20 and OC22, using the direct methods to predict forces has been shown to achieve better energy and force errors [1] and ultimately resulted in better IS2RE error, which is the most important metric on the two datasets. For MD17, we follow the practice of Equiformer and apply DeNS to energy-conserving forces.
>
> Reference:
>
> [1] https://pubs.acs.org/doi/full/10.1021/acscatal.2c02291

---

> > ### Comment · Reviewer_q8wm · 2024-11-06
> >
> > 1. With respect to MD-relevant benchmarks, I disagree with the authors claims that these are not established and not available for many models. In fact, the very popular paper https://arxiv.org/pdf/2210.07237 reports MD-relevant benchmarks for all of the models compared against in MD17 except for TorchMDNet. It would be entirely sufficient for the authors to perform this analysis only on MD17 for the EquiformerV2 model with and without DeNS and compare against the already available and established benchmark results, which should not be too much to ask. I believe these results are very important to show that the model and DeNS method may be actually useful for MD simulations.
> >
> > 2. I am happy with the changes made to expand upon the explanations for both equivariant models and the Equivformer models, I understand the authors reluctance to expand upon this in more detail so it doesn't detract from the main point of the paper which is the DeNS method, since it is model-agnostic.
> >
> > 3. I understand the authors justification for using/not using gradient-based forces based on the dataset and established practice/benchmarks for them, however I still think that the current text does not make this distinction clear enough. From the authors comment I was able to find the references to direct and gradient methods in section 3.1, but here it is only mentioned briefly and as an example. Again direct and gradient-based forces are mentioned in section 4.3, as the authors already pointed out, but the text does not make it clear enough that the gradient-based forces were used for MD17. It would be advisable and clearer to include the information about how the forces are obtained in the Dataset and Tasks subsections of sections 4.1., 4.2 and 4.3, as well as the respective training details sections in the supplement.
> >
> > If these changes I mentioned in points 1. and 3. are made I will be very happy to accept this work for publication.

---

> > > ### Author Response · Authors · 2024-11-12
> > > **Response to the comment by Reviewer q8wm on November 6th**
> > >
> > > Thanks for the feedback. We address the comments as below.
> > >
> > > ---
> > > > 8. [Reviewer Comment 1 on Nov. 6th] MD17 simulation results.
> > >
> > > Following the work [1], we run simulations on the four molecules (i.e., Aspirin, Ethanol, Naphthalene and Salicylic Acid) and compare the two simulation-based metrics, which are stability and distribution of interatomic distances h(r). We use the previously trained Equiformer for the simulations. We note that Equiformer models are trained on 950 examples for each molecule instead of 9,500 as in [1]. The results are summarized below. Training with DeNS helps energy and forces MAE as well as stability. The results of h(r) are similar. For Naphthalene, training with DeNS improves stability from 133.8/300 to 157.2/300. Both models become unstable when running simulations of Naphthalene, and we surmise that is because we only use 950 examples for training instead of 9,500. For others, the performance gain in simulation-based metrics is not significant since the original Equiformer trained on 950 examples already achieves similar results to other top-performing models trained on 9,500 examples, suggesting that the potential room for improvement would be quite limited.
> > >
> > > |                                 | Aspirin |        |           |      | Ethanol |        |           |      | Naphthalene |        |           |      | Salicylic Acid |        |           |      |
> > > |---------------------------------|---------|--------|-----------|------|---------|--------|-----------|------|-------------|--------|-----------|------|----------------|--------|-----------|------|
> > > |                                 | energy  | forces | stability | h(r) | energy  | forces | stability | h(r) | energy      | forces | stability | h(r) | energy         | forces | stability | h(r) |
> > > | Equiformer (L_{max} = 2)        | 5.3     | 7.2    | 300       | 0.02 | 2.2     | 3.1    | 289.9     | 0.09 | 3.7         | 2.1    | 133.8     | 0.12 | 4.5            | 4.1    | 300       | 0.03 |
> > > | Equiformer (L_{max} = 2) + DeNS | 5.1     | 5.7    | 300       | 0.02 | 2.2     | 2.6    | 300       | 0.09 | 3.7         | 1.7    | 157.2     | 0.12 | 4.4            | 3.7    | 300       | 0.03 |
> > >
> > > Reference:
> > > [1] Fu et al. Forces are not Enough: Benchmark and Critical Evaluation for Machine Learning Force Fields with Molecular Simulations. TMLR 2023.
> > >
> > > ---
> > > > 9. [Reviewer Comment 3 on Nov. 6th] It would be advisable and clearer to include the information about how the forces are obtained in the Dataset and Tasks subsections of sections 4.1., 4.2 and 4.3, as well as the respective training details sections in the supplement.
> > >
> > > We have added the details of whether forces are obtained with direct or gradient methods. Thanks for the concrete suggestion.

---

> > > > ### Comment · Reviewer_q8wm · 2024-11-12
> > > >
> > > > The authors have done great to include the additional evaluation in terms of MD-relevant metrics, as well as more clearly state when gradient methods have been used to obtain the forces. Given these changes I am quite happy with the current state of the manuscript.

---

### Decision · Action_Editor_4iVe · 2024-12-03

**Recommendation:** Accept as is

**Comment:**

The reviewers all agree that this work has value, it provides support for the claims and is interesting for the community.

**Audience:**

Yes

**Claims And Evidence:**

Yes